# The enteric pathogen *Cryptosporidium parvum* exports proteins into the cytosol of the infected host cell

Jennifer E Dumaine[1], Adam Sateriale[1†], Alexis R Gibson[1], Amita G Reddy[2], Jodi A Gullicksrud[1], Emma N Hunter[1], Joseph T Clark[1], Boris Striepen[1]*

[1]Department of Pathobiology, School of Veterinary Medicine, University of Pennsylvania, Philadelphia, United States; [2]Franklin College of Arts and Science, University of Georgia, Athens, United States

**\*For correspondence:**
striepen@upenn.edu

**Present address:** †The Francis Crick Institute, London, United Kingdom

**Competing interest:** The authors declare that no competing interests exist.

**Abstract** The parasite *Cryptosporidium* is responsible for diarrheal disease in young children causing death, malnutrition, and growth delay. *Cryptosporidium* invades enterocytes where it develops in a unique intracellular niche. Infected cells exhibit profound changes in morphology, physiology, and transcriptional activity. How the parasite effects these changes is poorly understood. We explored the localization of highly polymorphic proteins and found members of the *Cryptosporidium parvum* MEDLE protein family to be translocated into the cytosol of infected cells. All intracellular life stages engage in this export, which occurs after completion of invasion. Mutational studies defined an N-terminal host-targeting motif and demonstrated proteolytic processing at a specific leucine residue. Direct expression of MEDLE2 in mammalian cells triggered an ER stress response, which was also observed during infection. Taken together, our studies reveal the presence of a *Cryptosporidium* secretion system capable of delivering parasite proteins into the infected enterocyte.

## Editor's evaluation

Apicomplexa parasites export proteins into their infected cells to modulate/co-opt signaling pathways for their intracellular development. This study demonstrates for the first time the export of a protein, MEDLE2, from *Cryptosporidium parvum*, and characterizes its targeting signal as well as its effector function in the infected host cell.

## Introduction

The Apicomplexan parasite *Cryptosporidium* is a leading cause of diarrheal disease worldwide. Young children are highly susceptible to infection, and cryptosporidiosis is an important contributor to child mortality (*Khalil et al., 2018*; *Kotloff et al., 2013*). Children in resource-poor settings carry a disproportionate burden of severe disease (*Choy and Huston, 2020*). Malnutrition enhances the risk of severe cryptosporidiosis, and at the same time, the disease impacts the nutritional state of children, which can lead to impaired growth (*Costa et al., 2011*; *Mondal et al., 2009*). Infection with the parasite results in protective immunity, but this immunity is not sterile and may require multiple exposures to develop (*Chappell et al., 1999*; *Okhuysen et al., 1998*). Most human diseases are due to infection with *Cryptosporidium hominis,* which only infects humans, and *Cryptosporidium parvum,* which can be zoonotically transmitted (*Feng et al., 2018*; *Nader et al., 2019*). The emergence of *Cryptosporidium* species is driven by host adaptation resulting in specialization and narrowing host specificity;

however, the sexual life cycle of the parasite allows for recombination and can lead to rapid convergent evolution of host specificity (*Guo et al., 2015*; *Nader et al., 2019*).

*Cryptosporidium* infects the epithelium of the small intestine, where it lives in a unique intracellular, but extracytoplasmic niche (*Elliott and Clark, 2000*). The mechanism by which this niche is established during invasion is still debated but involves the rearrangement of the host actin cytoskeleton, the formation of tight junction-like structures between host and parasite membranes, and a dense band of unknown composition at the host-parasite interphase (*Bonnin et al., 1999*; *Elliott and Clark, 2000*). *Cryptosporidium* has severely reduced metabolic capabilities and relies heavily upon the host cell for nutrients and metabolites (*Abrahamsen et al., 2004*; *Xu et al., 2004*). A number of specialized uptake mechanisms have been proposed to fill this need, many of which are believed to be localized to the so-called feeder organelle at the host-parasite interface (*Guérin and Striepen, 2020*). In summary, *Cryptosporidium* remodels the host cell in significant ways that include its cytoskeleton (*Bonnin et al., 1999*; *Elliott and Clark, 2000*), cellular physiology and metabolism (*Argenzio et al., 1990*; *Kumar et al., 2018*), as well as aspects of immune restriction and regulation (*Laurent and Lacroix-Lamandé, 2017*).

Many bacterial, protozoan, and fungal pathogens use translocated effectors to manipulate their hosts to secure nutrients and to block host immunity. In *Plasmodium falciparum*, exported effectors form adhesive structures on the surface of red blood cells to alter tissue distribution and mechanical properties to prevent clearance (*Crabb et al., 1997*; *Leech et al., 1984*) and install new nutrient and ion uptake mechanisms (*Baumeister et al., 2006*). In *Toxoplasma gondii*, translocated effectors disarm critical elements of interferon-induced cellular restriction (*Gay et al., 2016*; *Olias et al., 2016*), establish access to cellular nutrients, and rewire signaling and transcriptional networks to antagonize immune responses to promote a favorable environment for parasite growth (*Bougdour et al., 2013*; *Braun et al., 2013*; *Braun et al., 2019*; *Gold et al., 2015*). *Cryptosporidium* has been hypothesized to use exported effector proteins to ensure its survival (*Pellé et al., 2015*); however, to date, no such factors have been identified. Here, we report that the polymorphic protein MEDLE2 is exported to the host cell cytoplasm by all stages of the *C. parvum* life cycle in vitro and in vivo. This protein is not injected during invasion, but rather is transported into the host cell following the initial establishment of infection. We carefully mapped the requirements for export and found a signal that includes a proteolytic cleavage site, and we demonstrate that exported proteins undergo processing. Overall, we demonstrate the presence of a robust translocation mechanism established by intracellular parasites that delivers parasite proteins to the host cell.

## Results

### The *C. parvum* protein MEDLE2 is exported into the host cell

The genome of *C. parvum* encodes multiple families of paralogous proteins that carry N-terminal signal peptides and share conserved amino acid repeat motifs among family members (*Abrahamsen et al., 2004*). These genes are often found within clusters (homologous and heterologous), many of which are located proximal to the telomeres of multiple chromosomes (*Figure 1A*). Comparing strains and species, these genes are highly polymorphic and vary in copy number, which has been interpreted as a sign of rapid evolution driven by their roles in invasion, pathogenesis, and host cell specificity (*Guo et al., 2015*; *Nader et al., 2019*; *Xu et al., 2019*) We hypothesized that such roles might be reflected in the targeting of presumptive effectors to the host cell and selected representatives from each polymorphic gene family for initial localization studies (*Table 1*). Selected loci were modified in the *C. parvum* IOWAII isolate using CRISPR/Cas9-driven homologous recombination (*Vinayak et al., 2015*) to append three hemagglutinin epitopes (3XHA) in translational fusion to the C-terminus (*Figure 1A*). Drug-resistant parasites were recovered for four of six initial candidates, and successful genomic insertion was mapped by PCR (*Figure 1B and C*, *Figure 1—figure supplement 1*). We next infected human ileocecal colorectal adenocarcinoma cell cultures (HCT-8) with transgenic parasites and assessed the localization of the tagged proteins by immunofluorescence assay (IFA, *Figure 1D and E*). For most candidates, the tagged protein (red) appeared to coincide with the parasite and/or the parasitophorous vacuole (cgd8_3560, *Figure 1D*, *Figure 1—figure supplement 1*). In contrast, upon infection with parasites tagged in the MEDLE2 locus (cgd5_4590), HCT-8 cells showed HA staining in the cytosol (*Figure 1E*). We note that the cells that stain for HA (red) were those that were

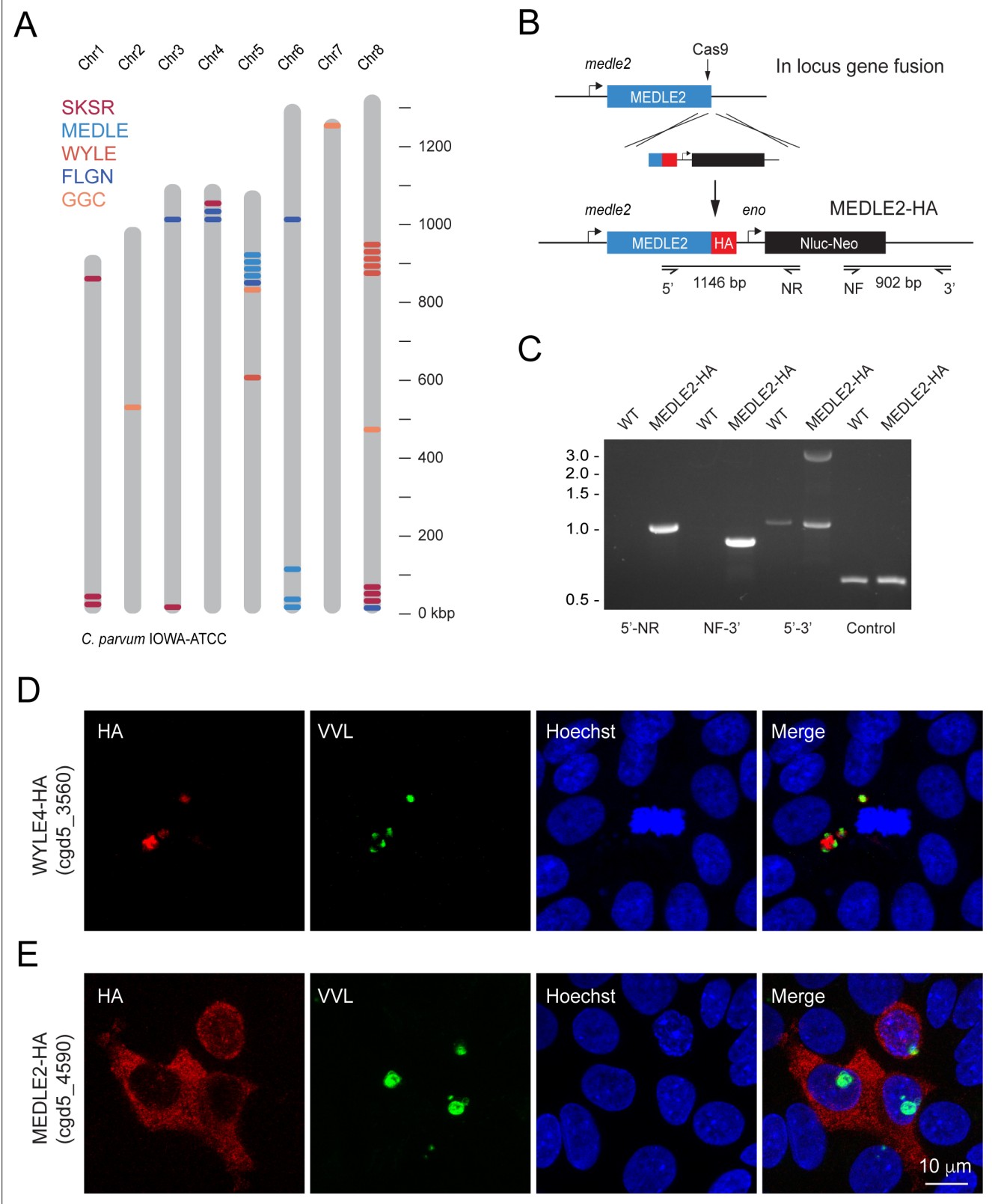

**Figure 1.** MEDLE2 is exported to the host cell cytoplasm. (**A**) Schematic overview of the chromosomal location for polymorphic gene families in the *C. parvum* genome. (**B**) Map of the MEDLE2 locus targeted in *C. parvum* for insertion of a 3× hemagglutinin (HA) epitope tag, a nanoluciferase reporter gene (Nluc), and neomycin phosphotransferase selection marker (Neo). (**C**) PCR mapping of the MEDLE2 locus using genomic DNA from wild type (WT) and transgenic (MEDLE2-HA) sporozoites, corresponding primer pairs are shown in (**B**), and thymidine kinase (TK) gene used as a control. Note

*Figure 1 continued on next page*

*Figure 1 continued*

the presence of two bands in the 5′–3′ amplification, indicating the presence of a transgene (3081 bp) and persistence of an unmodified copy (1174 bp), suggesting multiple copies of MEDLE2 in the *C. parvum* genome; also see *Figure 1—figure supplement 2*. (**D, E**) HCT-8 cultures were infected with WYLE4-HA (**D**) or MEDLE2-HA (**E**) transgenic parasites and fixed after 24 hr for immunofluorescence assay (IFA). Red, antibody to HA; green, *Vicia villosa* lectin stain, VVL (*Gut and Nelson, 1999*); blue, Hoechst DNA dye. Additional genes targeted and the localizations of their products are summarized in Table 1 and *Figure 1—figure supplements 1 and 3*.

The online version of this article includes the following figure supplement(s) for figure 1:

**Figure supplement 1.** Additional secretory proteins tested in this study.

**Figure supplement 2.** Knockout of MEDLE2 reveals multiple copies of the gene in the genome.

**Figure supplement 3.** Other members of the MEDLE gene family are exported to the host cell.

infected with parasites, labeled with *Vicia villosa* lectin (VVL, green), and conclude that MEDLE2-HA is exported by the parasite into the host cell during or following invasion.

We investigated whether other members of the MEDLE gene family are similarly exported and selected MEDLE1 (cgd5_4580) and MEDLE6 (cgd6_5490) for epitope tagging. In IFAs, both tagged MEDLE1 and MEDLE6 localized around the parasite, as well as to the host cell cytoplasm, but expression and export was less than what was observed for MEDLE2 (*Figure 1—figure supplement 3*). To explore this difference, we engineered a chimeric mutant carrying an extra copy of MEDLE1-HA driven by the MEDLE2 promoter. Upon infection with these parasites, we observed increased MEDLE1 expression making export into the host cell easier to appreciate (*Figure 1—figure supplement 3*). We conclude that multiple members of the MEDLE family are host-targeted proteins. Of these, MEDLE2 was expressed and exported most robustly and was selected for further study.

## MEDLE2 is exported across the *C. parvum* life cycle in culture and infected animals

To better understand export of MEDLE2 to the host cell, we next engineered a reporter parasite in which the endogenous locus of MEDLE2 was 3XHA epitope tagged, and these parasites expressed a tandem mNeon green fluorescent protein in their cytoplasm to allow for identification and quantification of parasites (MEDLE2-HA-tdNeon, *Figure 2—figure supplement 1*). These parasites were then used in time-course experiments across the 72 hr of infection afforded by the HCT-8 culture model. Following inoculation with sporozoites, cultures were fixed in 12 hr increments and processed for IFA. MEDLE2 was observed at all time points (*Figure 2A*) and quantification showed that the number of HA positive host cells (*Figure 2B*, red) increased over time, closely matching the increase in the number of parasites (blue). 94 ± 1.83% (mean ± SD, n = 3695) of the cells showing HA staining also showed parasite infection. Importantly, this high level of correlation between host cell HA expression and infection ($r^2$ = 0.9) remained constant over 72 hr. Previous work has shown synchronous life cycle progression over this time span. Initially, all parasites represent asexual stages replicating by merogony followed by a dramatic development switch at 40 hr, and later cultures are dominated by male and female gametes (*Tandel et al., 2019*). We staged parasites at 48 hr and identified female gametes and male gamonts using antibodies for the markers COWP1 and tubulin, respectively. We found the host cells infected with all these stages to be positive for MEDLE2-HA (*Figure 2D*).

We also tested whether MEDLE2 export occurs in vivo. Susceptible *Ifng*[-/-] mice were infected with 10,000 oocysts of the reporter strain, and after 12 days, mice were euthanized, the ileum was resected, fixed, and processed for histology. As shown by immunohistochemistry of sections of infected intestines in *Figure 2E*, MEDLE2 was exported to infected cells in vivo and exhibited cytoplasmic localization.

**Table 1.** Members of multigene families for which localization of protein product was initially attempted in this study.

| Gene family | Gene ID | Result |
|---|---|---|
| MEDLE | cgd5_4590 | Exported to host cell |
| FLGN | cgd4_4470 | Transgenic unsuccessful |
| GGC | cgd5_3570 | Transgenic unsuccessful |
| SKSR | cgd8_30 | Not in host cell |
| SKSR | cgd8_40 | Not in host cell |
| WYLE | cgd8_3560 | Not in host cell |

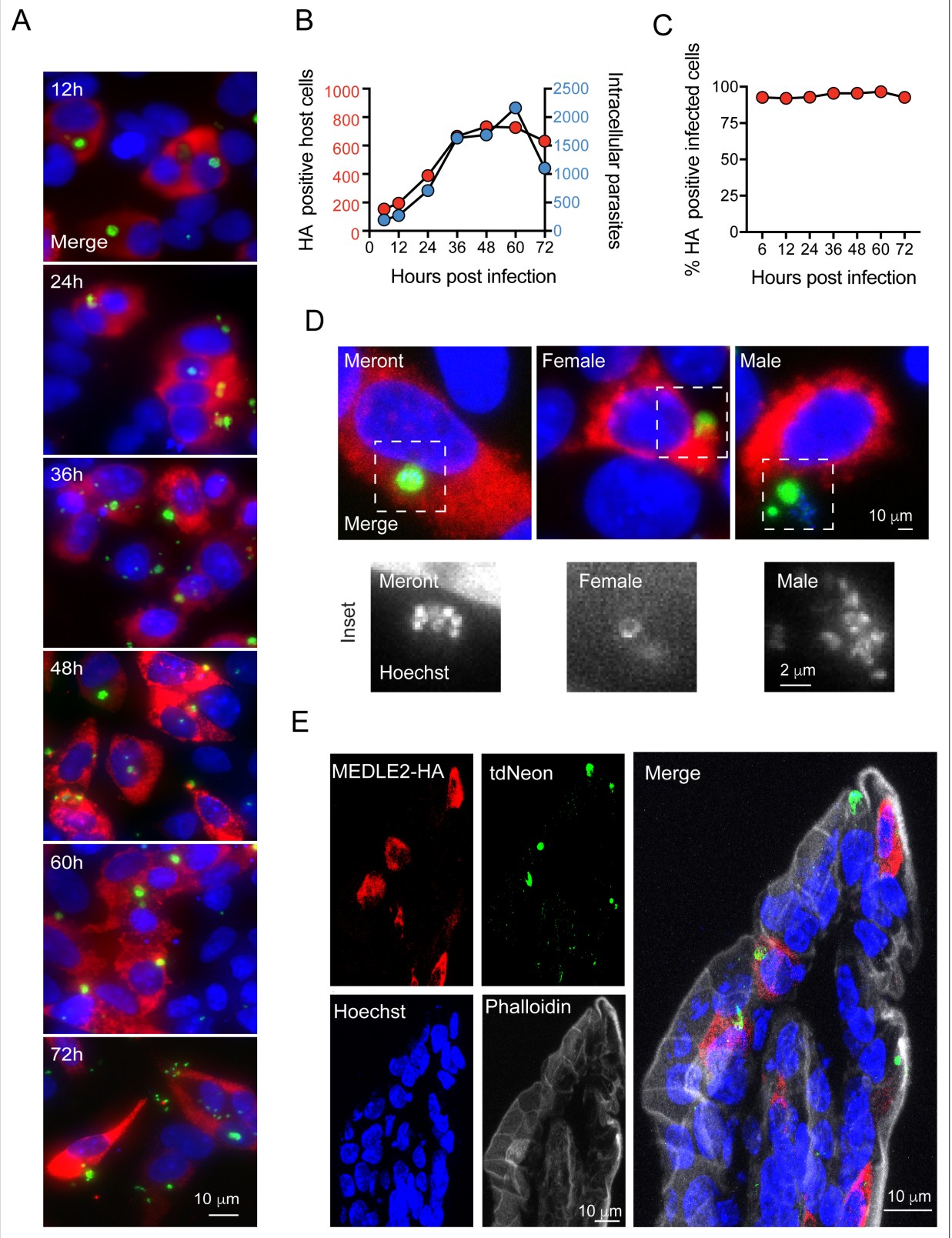

**Figure 2.** Infected cells express MEDLE2-HA across the parasite life cycle. (**A**) 200,000 MEDLE2-HA-tdNeon transgenic parasites were used to infect HCT-8 cells and fixed at intervals across a 72 hr time period. Data shown are representative images from triplicate coverslips processed for immunofluorescence assay (IFA). Red, hemagglutinin (HA)-tagged protein; green, parasites (mNeon); blue, Hoechst. (**B, C**) Quantification of MEDLE2-expressing cells (red) versus intracellular parasites (blue) for 3695 host cells evaluated across a 72 hr time course. 20 fields of view quantified using

*Figure 2 continued on next page*

*Figure 2 continued*

ImageJ to identify host cells and parasites (**B**). The percentage of cell exhibiting MEDLE2-HA and mNeon staining is constant across the time course with a cumulative 94 ± 1.83% (mean ± SD) (**C**). (**D**) HCT-8 cultures infected with MEDLE2-HA parasites were fixed for IFA at 48 hr when sexual life stages were present. Cells were stained with stage-specific antibodies for female (COWP1) and male (α- tubulin), demonstrating MEDLE2 is exported across the parasite life cycle. Red, HA-tagged protein; green, parasites (stage-specific antibody); blue, Hoechst. (**E**) IFA of cryosections from the small intestine of *Ifng*⁻/⁻ mice infected with MEDLE2-HA-tdNeon *C. parvum* (images representative of samples from three mice). Red, HA-tagged protein; green, parasites (tdNeon); blue, Hoechst; gray, Phalloidin (actin).

The online version of this article includes the following source data and figure supplement(s) for figure 2:

**Source data 1.** Numerical data used for the quantification of HA positive host cells and the intracellular parasites.

**Figure supplement 1.** Construction of a MEDLE2-HA cytoplasmic tdNeon reporter parasite.

## MEDLE2 is expressed and exported by trophozoites once infection has been established

Two broadly conserved temporal patterns have been described for host-targeted proteins of Apicomplexa (*Figure 3A*). Those involved in early aspects of the infection are packaged into the rhoptry organelle and injected during invasion (*Rastogi et al., 2019*). A second wave of proteins is translated and exported after the parasite has established its intracellular niche, and they are delivered to the host cell by a translocon-based mechanism (*de Koning-Ward et al., 2009*; *Franco et al., 2016*). For this reason, we next determined the timing of MEDLE2 expression using IFA of wild type (WT) and transgenic sporozoites mounted to cover glass with poly-lysine. We readily observed labeling for the sporozoite antigen Cp23 but did not detect any HA staining in transgenic sporozoites, indicating that MEDLE2 is not prepackaged into secretory organelles (*Figure 3B*). We then stained intracellular stages at different time points following invasion to determine the kinetics of MEDLE2 expression and export. At 4 hr, HA is first detectable, with labeling being associated with the parasite (parasite nuclei stained with Hoechst highlighted by white arrowheads). Beginning at 5.5 hr, MEDLE2-HA staining was observed throughout the host cell cytoplasm and continued to accumulate over time (*Figure 3C*).

MEDLE2 is predicted to be a 209 amino acid protein with a putative N-terminal signal peptide suggesting trafficking through the parasite's secretory pathway. To test this, we used brefeldin A (BFA), which blocks passage through the secretory pathway between the ER and Golgi. Cells were infected with MEDLE2-HA parasites, treated with BFA beginning 3 hr post infection, and fixed for IFA at 10 hr to assess MEDLE2 localization. BFA treatment initiated after invasion ablated export and resulted in the accumulation of MEDLE2-HA within the parasite (*Figure 3D*). Image analysis and quantification showed this reduction in export to be significant when comparing treated (red) to untreated cells (blue) (n = 98 treated; n = 198 untreated; p<0.0001; unpaired *t* test with Welch's correction; *Figure 3E*). We conclude that MEDLE2 is not injected by the sporozoite during invasion; rather, it is expressed and exported by the trophozoite and arrives in the host cell about 5 hr post infection (note that the length of the intracellular lytic cycle of asexual stages is 11.5 hr; *Guérin et al., 2021*). BFA does block parasite growth, and that treatment was therefore restricted to a single merogony cycle.

## MEDLE2 is an intrinsically disordered protein, and its export is blocked by ordered reporters

To further investigate the export of MEDLE2, we sought to develop a reporter assay to follow trafficking using three previously established systems. We fused the fluorescent reporter mScarlet to the C-terminus of MEDLE2 in its endogenous locus (*Figure 4A*, *Figure 4—figure supplement 1*). Transgenic parasites showed robust expression of the reporter when used to infect HCT-8 cultures. However, this fluorescence (red) was associated with parasites, highlighted by staining with VVL (*Figure 4B*, green), suggesting that MEDLE2-mScarlet remained trapped within or close to the parasite. We considered that export may occur, but that we lack the sensitivity to detect it. Thus, we engineered two highly sensitive assays that employ enzymes to amplify the signal.

First, we tagged MEDLE2 with beta-lactamase, which has been used to reveal effector export in bacteria and protozoa (*Charpentier and Oswald, 2004*; *Lodoen et al., 2010*). These parasites also expressed a cytoplasmic red fluorescent protein (*Figure 4—figure supplement 1B*). Cells infected with MEDLE2-BLA sporozoites were incubated with CCF4-AM, a cell-permeable substrate of beta-lactamase and imaged by live microscopy. Infected and uninfected cells accumulated CCF4-AM

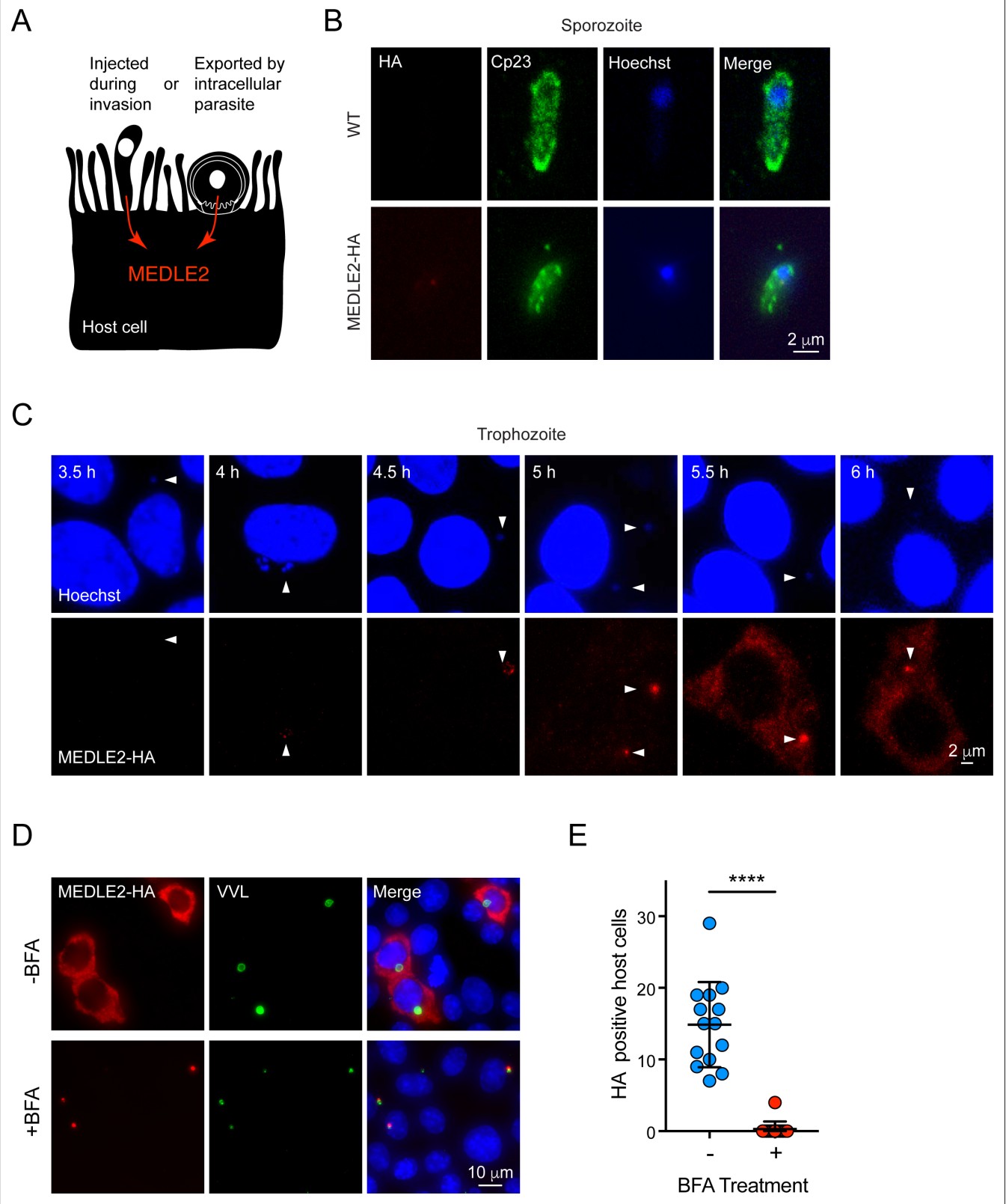

**Figure 3.** MEDLE2 is expressed by trophozoites and passes through the secretory pathway. (**A**) Schematic representation of hypothetical patterns of MEDLE2 export in *C. parvum*. (**B**) Immunofluorescence assay (IFA) of wild type (WT) and MEDLE2-HA sporozoites fixed on poly-L-lysine-treated coverslips. We note that MEDLE2-HA is not observed in sporozoites. Red, hemagglutinin (HA)-tagged protein; green, sporozoite antigen Cp23; blue, Hoechst. (**C**) HCT-8 cells infected with MEDLE2-HA parasites were fixed in 30 min increments and processed for IFA. Data shown are representative

*Figure 3 continued on next page*

*Figure 3 continued*

images from a time-course bridging 3 hr (no observed MEDLE2-HA) and 6 hr (MEDLE2-HA abundant in host cell). White arrowheads denote parasite nuclei. Red, HA-tagged protein; blue, Hoechst. (**D**) MEDLE2-HA parasites were excysted and used to infect HCT-8 and after 3 hr media were supplemented with brefeldin A (BFA) (10 µg/mL). 10 hr post infection, cells were fixed and processed for IFA. Red, HA-tagged protein; green, parasites (VVL); blue, Hoechst (**D**). (**E**) The impact of BFA treatment on MEDLE2-HA export was quantified showing a significant reduction in MEDLE2 export when comparing BFA-treated (red) and untreated cells (blue) (n = 191 untreated, n = 98 treated; mean ± SEM; p<0.0001; unpaired *t* test with Welch's correction).

The online version of this article includes the following source data for figure 3:

**Source data 1.** Numerical data used for the quantification of HA positive host cells in the absence and presence of Brefeldin A (BFA).

(***Figure 4C***, green); however, we did not detect substrate cleavage, resulting in blue fluorescence (***Figure 4C***). This could be due to the lack of MEDLE-BLA expression or export. To visualize localization of the MEDLE2-BLA fusion protein during infection, we performed IFA on MEDLE2-BLA-infected cells using a BLA antibody and observed that MEDLE2-BLA (green) was expressed but remained with the parasite (***Figure 4—figure supplement 1C***, red). Next, we generated a MEDLE2 fusion with Cre recombinase and used these parasites to infect an HCT-8 host cell line we engineered to carry a floxed GFP/RFP color switch reporter (see ***Figure 4—figure supplement 2***). After 48 hr, cells were subjected to flow cytometry (***Figure 4D***). Using a gate for live, single cells, uninfected cells showed green (95.1%) but not red fluorescence (2.51%). Transfection of host cells with a Cre recombinase expression plasmid resulted in a pronounced shift to red fluorescence (+ control, 32%, ***Figure 4D***). Cells infected with either WT parasites or MEDLE2-Cre parasites remained green, despite robust infection (***Figure 4—figure supplement 2***). Therefore, we tested three reporters, none of which resulted in detectable export to the host cell. We note that multiple algorithms predict MEDLE2 to be a highly disordered protein (***Figure 4—figure supplement 3***; low-complexity regions are indicated in light blue in ***Figure 5B***) and conclude that translocation is blocked when folded reporters are fused to the protein.

## MEDLE2 export depends upon N-terminal sequence features

We then sought to determine whether MEDLE2 contains sequence-specific information for host targeting. Using previously established host-targeting motifs from *P. falciparum* and *T. gondii* as models (***Coffey et al., 2015***; ***Marti et al., 2004***), we searched the MEDLE2 amino acid sequence to identify candidate export motifs. Preference was given to regions with a basic amino acid, followed by one or two random amino acids, and a leucine residue (***Pellé et al., 2015***). While *Plasmodium* host-targeting motifs are typically found in close proximity to the signal peptide, *T. gondii* exhibits less rigid distance requirements (***Coffey et al., 2015***). We identified four motifs, three sites in proximity to the N-terminus and one C-terminal candidate for mutational analysis (***Figure 5A***). As folded reporters are not tolerated, we engineered parasite lines to express an ectopic copy of MEDLE2 marked by an HA tag. A cassette driven by the MEDLE2 promoter was inserted into the locus of the dispensable thymidine kinase gene (TK, ***Figure 5B***), and expression level and export of ectopic WT protein was indistinguishable from protein tagged within the native locus.

Removal of the sequence encoding the N-terminal signal peptide (ΔSP) prevented MEDLE2-HA export, and the resulting protein accumulated within the parasite (***Figure 5C***). Next, we constructed a series of parasite strains in which each of the candidate motifs was replaced by a matching number of alanine residues (all mutants were confirmed by PCR mapping and Sanger sequencing; ***Figure 5—figure supplement 1***). Mutagenesis of three of these candidate motifs had no impact on MEDLE2 translocation to the host cell (***Figure 5C***, ***Figure 5—figure supplement 1***). In contrast, when the most N-terminal sequence KDVSLI was changed to six alanines, HA staining accumulated in the parasite and host cell staining was lost (***Figure 5C***). We conclude that in this mutant MEDLE2-HA is made but export is ablated. Next, we constructed six additional strains using the same strategy to change each amino acid position of the KDVSLI motif to alanine 1 residue at a time (***Figure 5—figure supplement 1***). Mutation of residue leucine 35 to alanine (L35A) ablated export and instead MEDLE2-HA remained with the parasite (***Figure 5D***). Changing the remaining five amino acids individually did not alter MEDLE2 localization in the host cell (***Figure 5D***). We conclude leucine 35 to be critical for export.

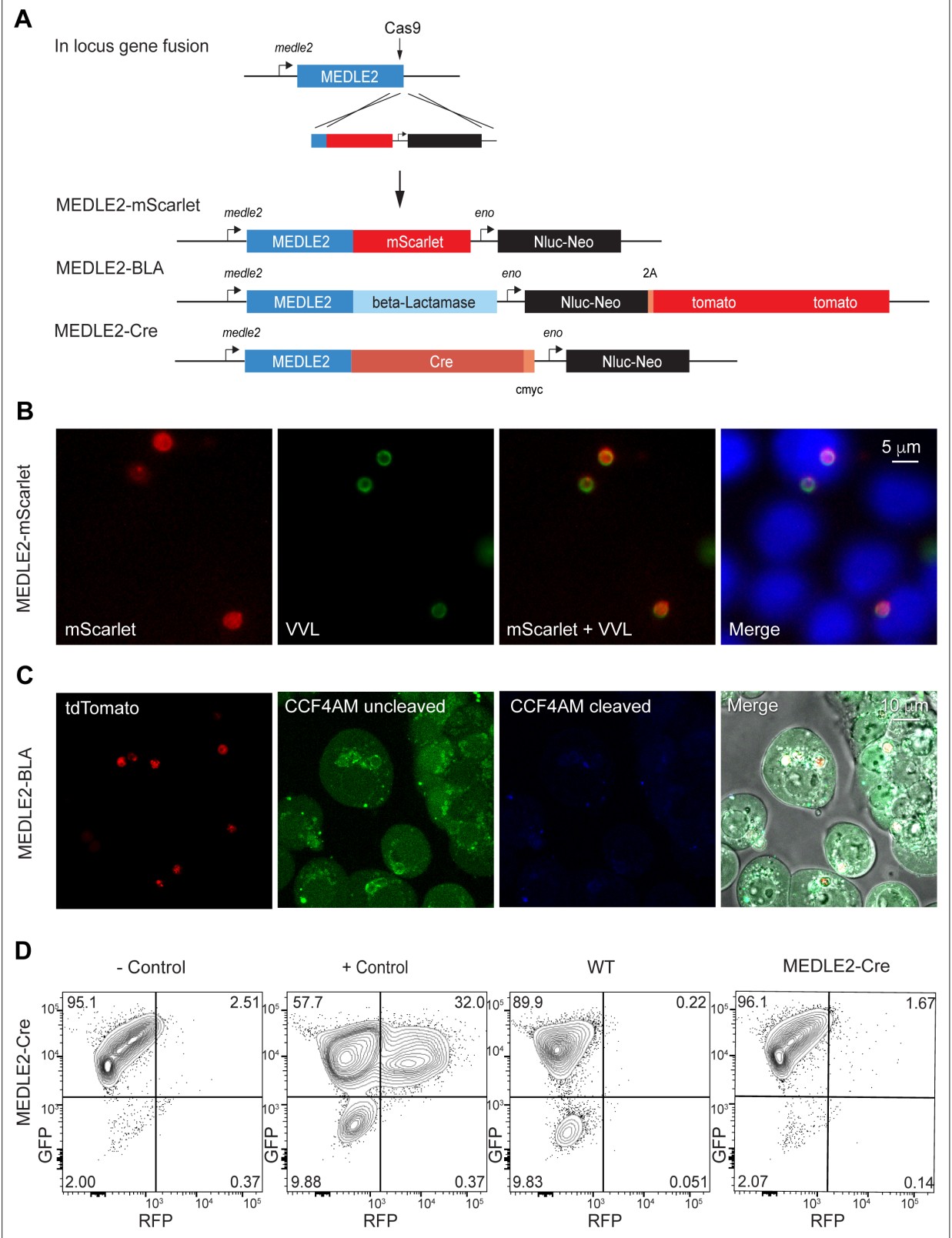

**Figure 4.** Ordered reporters disrupt MEDLE2 export. (**A**) Schematic map of the MEDLE2 locus targeted for insertion of three different reporter genes (mScarlet, beta-lactamase, or Cre recombinase), nanoluciferase (Nluc), and the selection marker (Neo). The guide RNA and flanking sequences used here were the same as those employed to generate MEDLE2-HA transgenic parasites (see *Figure 2—figure supplement 1*, *Figure 4—figure supplement 1* for more detail). (**B**) MEDLE2-mScarlet parasites were used to infect HCT-8 cells and fixed for immunofluorescence assay (IFA) across a

*Figure 4 continued on next page*

*Figure 4 continued*

time course. Data shown are from 10 hr post infection, which is representative of the MEDLE2 localization observed at all time points. Red, Medle2-mScarlet; green, parasites (VVL); blue, Hoechst. (**C**) HCT-8 cells were infected with MEDLE2-BLA *C. parvum* for 24 hr before incubation with the CCF4-AM beta-lactamase substrate and visualization by live microscopy. This experiment was repeated three times. Red, parasites (tdTomato); green, uncleaved CC4F-AM; blue, cleaved CCF4-AM; gray, DIC. We attribute lack of CCF4-AM cleavage to failure of MEDLE2-BLA to export (***Figure 4—figure supplement 1***). (**D**) MEDLE2-Cre parasites were used to infect loxGFP/RFP color switch HCT-8 cells ((***Figure 4—figure supplement 2***) for schematic representation). After 48 hr, cells were subjected to flow cytometry. Live, single cells were gated based upon forward and side scatter, and green fluorescence (GFP) and red fluorescence (RFP) were measured to detect Cre recombinase activity. Despite robust infection, MEDLE2-Cre-infected cultures did not express RFP (***Figure 4—figure supplement 2***) compared to the positive control that was transiently transfected to express Cre recombinase.

The online version of this article includes the following figure supplement(s) for figure 4:

**Figure supplement 1.** MEDLE2-BLA is not exported.

**Figure supplement 2.** MEDLE2-Cre parasites infect loxGFP/RFP color switch cells.

**Figure supplement 3.** MEDLE2 is an intrinsically disordered protein.

## MEDLE2 export is linked to proteolytic processing

For both *P. falciparum* and *T. gondii*, leucine residues serve as crucial sites for proteolytic processing events that license proteins to leave the parasite and enter the host cell (***Coffey et al., 2015***; ***Hiller et al., 2004***; ***Marti et al., 2004***). To test whether such processing occurs during export of MEDLE2 in *C. parvum*, we performed western blot analysis on whole-cell lysates of infected cell cultures using antibodies to HA (MEDLE2-HA, green) and the drug resistance marker Neo (loading control, red). For WT MEDLE2-HA, we observed a single band with an apparent molecular weight of 31 kDa (***Figure 5E***, the predicted molecular weight for full-length MEDLE2-HA is 26.9 kDa but the abundance of positive charges is likely to result in reduced electrophoretic mobility). Protein KPVLKN/6A, carrying a mutation that did not affect trafficking to the host cell cytoplasm, was of identical size to WT MEDLE2-HA. In contrast, the KDVSLI/6A mutant, which is no longer exported, appeared to be of a larger molecular weight (***Figure 5E***). The mutant lacking the 22 amino acid signal peptide (ΔSP) produced an intermediate size band larger than the exported WT but smaller than the retained ΔKDVSLI mutant. We found a very similar pattern when analyzing the single amino acid mutants, where the L35A change caused the mutant protein to migrate more slowly when compared to WT or the ΔSP mutant (***Figure 5F***). To ensure the observed differences in apparent molecular weight were due to processing by the parasite and not the consequence of folding or subsequent host processing, we also expressed WT and mutants in mammalian cells (see Materials and methods for detail). In this context, the proteins are the same size (***Figure 5—figure supplement 2***). Overall, we interpret the relative sizes of the mutated proteins to indicate processing of MEDLE2 at a point beyond the signal peptide, a position that would be consistent with leucine 35. We note that this processing appears to require translocation into the ER as it does not occur in mutants lacking a signal peptide. Furthermore, the L35A mutation apparently prevented removal of the signal peptide, suggesting that processing at L35 could replace the canonical signal peptidase activity.

## MEDLE2 induces an ER stress response in the host cell

To begin to understand the consequence of MEDLE2 export on the host cell, we expressed the protein in human cells. MEDLE2 omitting the N-terminal signal peptide (aa 2–20) was codon optimized for human cell expression, appended to the N-terminus of GFP, and the resulting plasmid introduced into HEK293T cells by Lipofectamine transfection (we note that this protein contains 15 amino acids [aa 21–35] that are likely missing in the parasite exported protein due to N-terminal processing). A GFP-only parent plasmid served as control for the impact of transfection on cellular responses. Transfection with both constructs resulted in cytoplasmatic green fluorescence in roughly 40% of cells 24 hr post transfection (***Figure 6A***). GFP-positive cells were enriched by flow cytometry and the resulting populations were subjected to mRNA sequencing (three biological repeats for each sample, ***Figure 6B***). Differential gene expression analysis revealed 413 upregulated genes and 487 genes with lower transcript abundance in MEDLE2-GFP-expressing cells compared to cells expressing GFP alone, with an adjusted *p*-value less than 0.05 (***Figure 6C***). Gene set enrichment analysis (GSEA) detected changes in multiple pathways following introduction of MEDLE2 to host cells

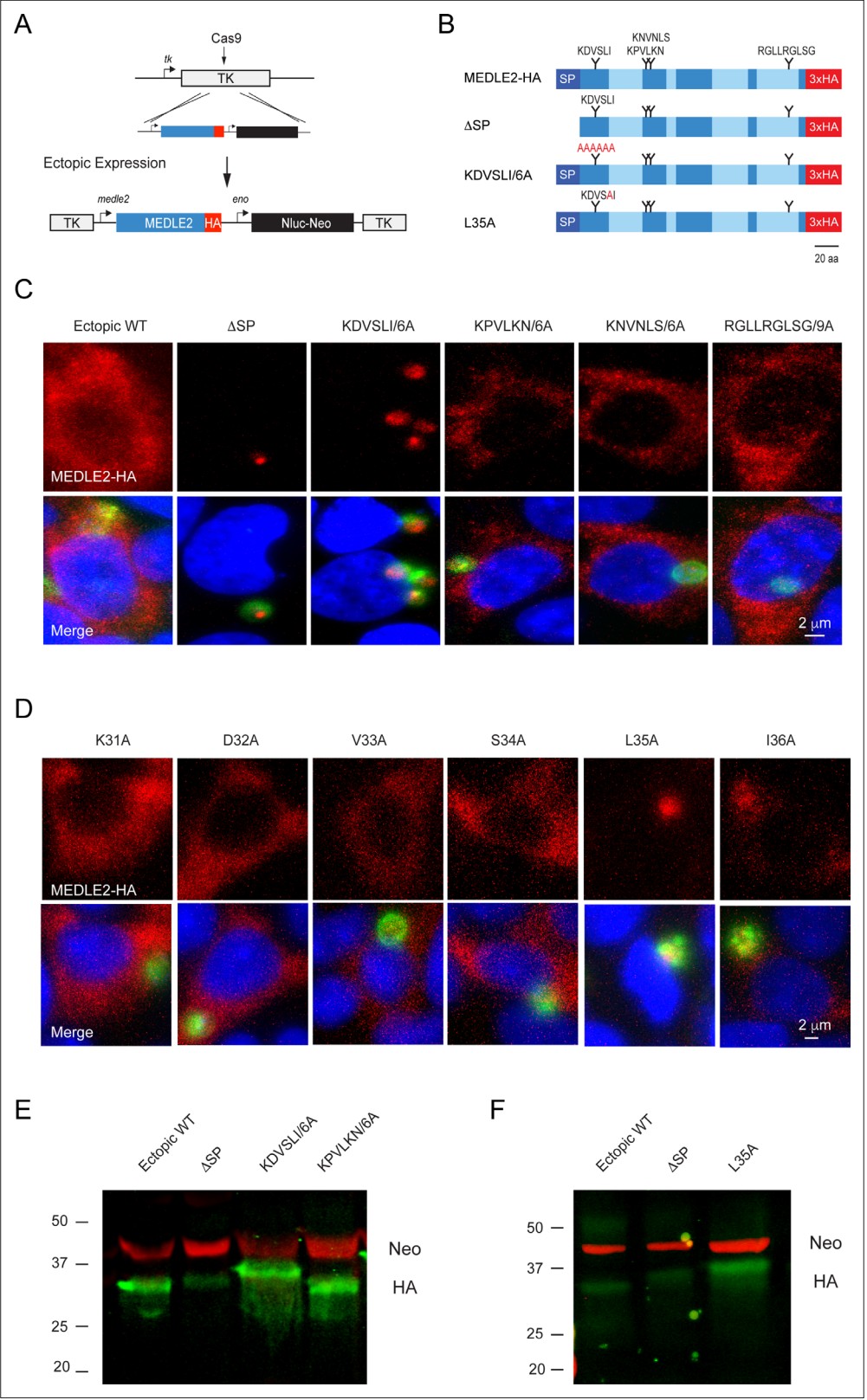

**Figure 5.** MEDLE2 contains a host-targeting motif that is processed during export. (**A**) Map showing the strategy used to engineer an ectopic copy of MEDLE2-HA in the thymidine kinase (TK) locus. Expression of an ectopic copy of MEDLE2-HA was driven by the MEDLE2 promoter. All point mutations were confirmed by Sanger sequencing (***Figure 5—figure supplement 1***). (**B**) Schematic representation of the MEDLE2 mutants generated using the

*Figure 5 continued on next page*

*Figure 5 continued*

strategy outlined in (**A**). The signal peptide (SP) is represented by dark blue, and low-complexity regions are shown in light blue. Candidate motifs targeted for mutagenesis are indicated with black triangles, and mutagenized amino acids are shown in red for two representative mutants. (**C, D**) Mutant parasites were used to infect HCT-8 cells and fixed for immunofluorescence assay (IFA) after 24 hr. For mutants shown in (**C**), the entire candidate motif was replaced with a matching number of alanine residues (e.g., KDVSLI/6A → AAAAAA). For mutants shown in (**D**), each individual amino acid in the KDVSLI sequence was changed to alanine. Red, hemagglutinin (HA)-tagged protein; green, parasites (VVL); blue, Hoechst. We note that SP and leucine 35 within the KDVSLI sequence are required for MEDLE2 export. (**E, F**) $5 \times 10^6$ transgenic oocysts were used to infect HCT-8 cells for 48 hr before preparation of whole-cell lysates. Proteins were separated by for SDS-PAGE and analyzed by western blot. The resulting blots for infections with whole motif mutants (**E**) and individual amino acid point mutants (**F**) are shown. Red, neomycin; green, HA. Note that when mutants are expressed in mammalian cells and not *C. parvum* the resulting proteins do not show any size differences (***Figure 5—figure supplement 2***).

The online version of this article includes the following figure supplement(s) for figure 5:

**Figure supplement 1.** Sanger sequencing confirming the generation of MEDLE2 mutants.

**Figure supplement 2.** MEDLE2 mutants are of the same size as wild type (WT) MEDLE2 when expressed in HEK293T cells.

**Figure supplement 3.** Uncropped images of panels shown in ***Figure 5D*** including infected and uninfected cells.

(***Figure 6—source data 1***). Notably, many of these pathways converged upon translation, ribosomes, amino acid metabolism, and associated signaling (***Figure 6—source data 1***). Functional annotation clustering using the Database for Annotation, Visualization and Integrated Discovery (DAVID) of the GSEA results revealed multiple responses contributing to ER stress, including changes in genes linked to the unfolded protein response (UPR). Genes that are part of the core enrichment of the ER stress response are highlighted (red) in the volcano plot, and the most upregulated genes in the ER stress signaling pathway identified by GSEA are identified by name (***Figure 6D***). Genes that were differentially expressed at an adjusted *p*-value less than 0.01 were used to derive a MEDLE2 response signature from the transfected cell dataset (234 genes). We next sought to explore whether the MEDLE2 gene signature observed following cell transfection was also present during *Cryptosporidium* infection using a published mRNA sequencing dataset generated from *C. parvum* infection of a homeostatic mini-intestine model (***Nikolaev et al., 2020***). We found enrichment for this MEDLE2 response signature with 51 genes of the MEDLE2 response present, and 22 of them contributing to the core enrichment of this response (***Figure 6E***, ***Figure 6—source data 2***). As further validation that ER stress responses also occur during in vivo infection, we performed qPCR on ileal segments resected from mice infected with *C. parvum* or those that were uninfected. We measured the RNA abundance for the four genes highlighted in the volcano plot and found three to be upregulated in infected mice compared to uninfected controls (NUPR1, CHAC1, DDIT3, ***Figure 6F***).

We next wanted to explore whether the observed ER stress response impacts on parasite survival. HCT-8 cell cultures were treated with chemical inhibitors of the PERK (GSK2606414) and IRE1 (KIRA6) UPR signaling pathways and infected with nanoluciferase expressing *C. parvum* parasites. After 24 hr, cultures treated with KIRA6 showed reduced parasite growth when compared to the DMSO vehicle control (one-way ANOVA, Dunnett's multiple comparisons test p=0.0303). Thapsigargin and GSK2606414 treatment did not affect parasite growth (***Figure 6G***). To test whether manipulation of the ER stress signaling pathways impacts parasite growth in vivo, we infected *Ddit3-/-* mice with MEDLE2-HA *C. parvum* parasites and monitored fecal relative luminescence as a measure of parasite shedding. *C. parvum* infection of *Ddit3-/-* mice resulted in a 56% reduction in the area under the curve compared to infection in C57BL/6J control mice (3,189,123 ± 69,887, 1,416,227 ± 44,850; total peak area ± standard error; unpaired *t* test, p<0.001; ***Figure 6H***). Taken together, the ER stress response triggered by ectopic expression of MEDLE2 in mammalian cells is also observed during parasite infection in culture and mice and may contribute to parasite survival.

## Discussion

Intestinal cryptosporidiosis in animals and humans is caused by parasites that are morphologically indistinguishable, and therefore were initially described as a single taxon, *C. parvum*. Extensive

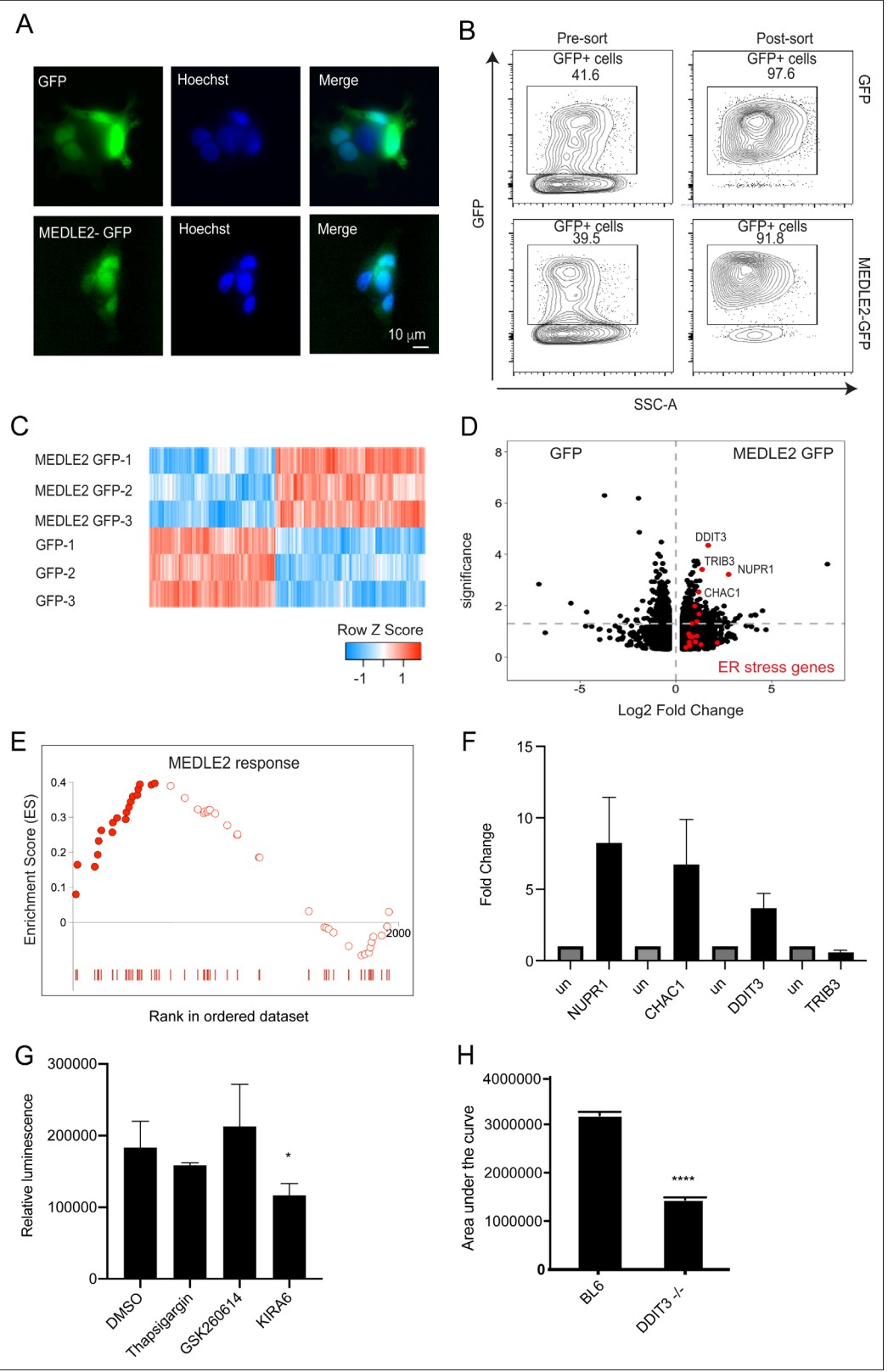

**Figure 6.** MEDLE2-expressing cells exhibit upregulation of genes involved in the unfolded protein response. (**A**) HEK293T cells were transfected with plasmids encoding MEDLE2-GFP or GFP alone. After 24 hr, cells were fixed and processed for immunofluorescence assay (IFA). GFP is shown in green, Hoechst in blue. (**B**) 24 hr post transection, HEK293T cells were trypsinized and double sorted for live, GFP+ singlets directly into RNA lysis buffer

*Figure 6 continued*

and subjected to RNA sequencing. (**C**) Heat map depicting the differential gene expression between MEDLE2-GFP (top panel) and GFP control expressing cells (bottom panel). Upregulated gene expression is shown in red (row Z score > 0), while blue shows genes that are downregulated in expression (row Z score < 0). Expressing cells compared to GFP control cells. 413 transcripts showed upregulation in MEDLE2-GFP-expressing cells (right) and 487 genes had lower transcript abundance (left). The horizontal dashed line indicates p-value = 0.05. Gene set enrichment analysis (GSEA) performed on the 900 differentially expressed genes from the MEDLE2 transfection dataset identifies core enrichment of 20 genes that belong to ER stress response signaling pathways, which are indicated on the volcano plot in red. The most upregulated genes are identified by their gene ID. (**E**) The 234 genes with the greatest differential expression (p<0.01, log fold change absolute value > 1.5) were used to define a MEDLE2 gene set from the MEDLE2-GFP transfection dataset. This signature was used to perform GSEA using data from single-cell RNA sequencing on *C. parvum*-infected organoid-derived cultures, which showed enrichment of 51 genes with 22 genes in the core enrichment for the MEDLE2 response set highlighted in solid red. We note that we did not detect the MEDLE2 response signature in datasets from other enteric infections including rotavirus (***Figure 6—figure supplement 1***). (**F**) Ileal sections were removed from *C. parvum*-infected *Ifng*[-/-] mice and uninfected controls (each n = 3), and expression levels for the four differentially upregulated genes in the MEDLE2 response set (NUPR1, CHAC1, DDIT3, and TRIB3) were measured by qPCR. (**G**) HCT-8 cultures were pretreated for 2 hr with inhibitors (GSK2606414 and KIRA6) of ER stress signaling pathways prior to infection with 10,000 MEDLE2-HA parasites. After 24 hr, cells were lysed and nanoluciferase assay was performed as a measure of parasite growth. Inhibition of the IRE1 signaling pathway with KIRA6 significantly reduced parasite growth (one-way ANOVA, Dunnett's multiple comparisons test p=0.0303; a representative experiment is shown [n = 6]). This experiment was repeated three times. (**H**) *Ddit3-/-* and C57BL/6J mice were treated with anti-mouse-IFN gamma antibody 1 day prior to infection with 10,000 MEDLE2-HA-tdNeon oocysts, and again at day 2 of infection. Fecal luminescence was determined by nanoluciferase activity to calculate the area under the curve for the duration of the infection. *Ddit3-/-* mice exhibited a 56% reduction in infection (1,416,227 ± 44,850; total peak area ± standard error) compared to control mice (3,189,123 ± 69,887; unpaired *t* test, p<0.001; n = 4 mice per group). One representative experiment is shown, which was repeated two more times, with a 54% reduction and no change in infection being observed.

The online version of this article includes the following source data and figure supplement(s) for figure 6:

**Source data 1.** Source data for ***Figure 6***.

**Source data 2.** Source data for ***Figure 6***.

**Figure supplement 1.** MEDLE2 response signature is absent from intestinal epithelial cells infected with human rotavirus.

**Figure supplement 2.** Transfection of HEK293T cells with *T. gondii* GRA16-GFP also results in ER stress.

population genetic studies have since replaced *C. parvum* with a multitude of species, subspecies, and strains (***Checkley et al., 2015***; ***Xiao et al., 1999***). The genomes of these parasites reflect the overall high degree of similarity in their conservation of gene content and synteny; nonetheless, these parasites show pronounced differences in their host specificity (***Feng et al., 2017***; ***Feng et al., 2018***; ***Nader et al., 2019***). The genomic differences observed are focused on families of predicted secretory proteins that have been proposed to contribute to host specificity, prominently among them the MEDLE family (***Fei et al., 2018***; ***Li et al., 2017***; ***Su et al., 2019***). Previous studies found that treatment of sporozoites with antisera raised against recombinant MEDLE1 or MEDLE2 diminished the efficiency of infection by 40%. This led the authors to consider that MEDLE proteins may play a role in invasion and that host specificity is rooted in receptor specificity (***Fei et al., 2018***; ***Li et al., 2017***). In this study, we screened polymorphic genes for the localization of the proteins they encode by epitope tagging the endogenous loci. We found that *C. parvum* exports multiple members of the MEDLE protein family into the cytosol of the host cell. This is particularly robust for the highly expressed MEDLE2 protein, where the observation is based on more than 15 independent transgenic strains using different epitope tags and antibodies, in locus and ectopic tagging, and held true in cultured cells and infected animals. Importantly, mutation of the tagged gene changed the localization and molecular weight of the protein, highlighting the specificity of the reagents used.

Apicomplexa evolved multiple mechanisms to deliver proteins to their host cells during and following invasion. The timing of MEDLE2 expression and export and its sensitivity to BFA treatment argues for a mechanism that becomes active after the parasite has established its intracellular niche, in contrast to injection during invasion from organelles already poised to secrete. In *P. falciparum,* such

export relies on the translocon complex PTEX (*de Koning-Ward et al., 2009*), which uses the ATPase activity of a heat shock protein to unfold and then extrude cargo through a transmembrane channel into the red blood cell (*Ho et al., 2018*; *Matthews et al., 2019*). Genetic ablation of this complex results in loss of export and loss of parasite viability (*Beck et al., 2014*; *Elsworth et al., 2014*). In *T. gondii,* proteins are exported during intracellular growth via the MYR complex, which is independent of ATP hydrolysis and requires intrinsically disordered cargo (*Hakimi et al., 2017*). Translocon and export are dispensable for growth of this parasite in culture, but loss profoundly impacts virulence in vivo (*Hakimi et al., 2017*; *Marino et al., 2018*). With the exception of a putative HSP101 chaperone, genome searches do not readily identify *C. parvum* homologs of the components that make up these two previously characterized translocons (*Pellé et al., 2015*), but nonetheless, there are important parallels that suggest conserved features of Apicomplexan export.

We demonstrate that MEDLE2 export depends on an N-terminal signal peptide and a leucine residue at position 35. Point mutation of this residue ablates export and results in a higher apparent molecular weight of the protein, which we interpret as a lack of processing at the mutated site. Disorder may be critical for MEDLE2 export as fusion of well-folded domains potently blocked its export into the host cell. This is consistent with an export mechanism unable to unfold proteins as proposed for *T. gondii* and *Plasmodium* liver stages (*Beck and Ho, 2021*; *Gehde et al., 2009*; *Marino et al., 2018*).

Export of proteins in *P. falciparum* and *T. gondii* similarly requires a host-targeting motif (*Coffey et al., 2015*; *Hiller et al., 2004*; *Marti et al., 2004*). While the sequence of the motifs varies between species, all share a required leucine residue that has been linked to processing by an aspartyl protease to license export (*Boddey et al., 2010*; *Coffey et al., 2015*; *Hammoudi et al., 2015*; *Russo et al., 2010*). This protease localizes to the ER in *P. falciparum* where it appears to replace the activity of signal peptidase for exported proteins, while the *T. gondii* homolog acts in the Golgi (*Coffey et al., 2015*; *Marapana et al., 2018*). Our mutational analysis in *C. parvum* suggests that cleavage of L35 could replace the activity of signal peptidase similar to *P. falciparum*. The *C. parvum* genome encodes six putative aspartyl proteases, three of which share significant similarity with the Plasmepsin V/ASP5 enzymes responsible for processing exported proteins in *P. falciparum* and *T. gondii*. Ablating these genes may provide an opportunity to block export to the host cell.

Pathogens inject a wide array of factors into the cells of their hosts to remodel cellular architecture, hijack death and survival pathways, change cell physiology and metabolism, and rewire transcriptional networks. Many of them aid immune evasion and the evolutionary arms race between host and pathogen and are potent modulators of host specificity (*Long et al., 2019*; *van der Does and Rep, 2007*). Here, we have identified the first examples of host-targeted effector proteins for intracellular stages of *C. parvum*. We introduced the protein into its target compartment, the mammalian cytosol, by transient transfection, and mRNA sequencing revealed a transcriptional signature of ER stress and UPR upon expression of MEDLE2, but not in matched control transfection with a GFP-only-expressing plasmid. We found that an ER stress response signature was also detectable when cells or mice were infected with *C. parvum*, which supports previous work identifying a UPR response during *C. parvum* infection in an in vitro model of infection (*Morada et al., 2013*).

ER stress is a common feature of intracellular infection that greatly modulates host cell survival and inflammatory response. Bacteria, viruses, and protozoa alike have evolved strategies to trigger or block the UPR during infection, suggesting the response not to be a by-product of infection, but rather an active participant (*Abhishek et al., 2018*; *Alshareef et al., 2021*; *Perera et al., 2017*). This relationship is complex, and depending on cellular context, the UPR can aid pathogens or restrict them. Modulating the UPR is pivotal for intracellular bacteria that reside in vacuoles, including *Chlamydia, Brucella, Listeria,* and *Mycobacterium,* to build their intracellular niche and to acquire nutrients (*Celli and Tsolis, 2015*). *Cryptosporidium* lost the ability to synthesize most required metabolites and depends on an elaborate membranous feeder organelle at the host-parasite interface. This reduction includes biosynthesis of many lipids (*Coppens, 2013*), and we note modulation of lipid synthesis and mobilization as one important consequence of the UPR (*Moncan et al., 2021*). The UPR is also important in the context of recognition of intracellular pathogens and the subsequent cytokine-mediated immune responses. *Cryptosporidium* triggers an enterocyte intrinsic inflammasome leading to the release of IL-18 from the infected cell (*McNair et al., 2018*), and infection also results in pronounced production of interferon $\lambda$ (*Ferguson et al., 2019*). We also tested GRA16,

a disordered effector protein of *T. gondii* in similar experiments and found it to similarly upregulate the ER stress response. This may indicate that such upregulation is inherent to the structure of these proteins (*Figure 6—figure supplement 2*). However, we note that manipulation of ER stress signaling using pharmacological inhibition of IRE1 in culture or genetic ablation of DDIT3 in mice resulted in reduced growth of *C. parvum*, suggesting that this response may contribute to parasite survival. We deem it likely that MEDLE2 has additional roles yet to be determined.

The MEDLE genes are highly polymorphic and encoded by multiple loci, but our current view of the family likely underestimates its true plasticity. The recent reannotation of the *C. parvum* genome using long read sequencing found evidence for multiple 'alternative' telomeres carrying different MEDLE gene clusters (*Baptista et al., 2021*). Our experimental observations support this. When we ablated the MEDLE2 gene by homologous recombination, additional MEDLE2 copies remained in the genome (*Figure 1—figure supplement 2*). Similar observations were made when tagging the gene (*Figure 1B*, *Figure 2—figure supplement 1*). It is important to note that *Cryptosporidium* has a single host life cycle and undergoes sexual exchange throughout the infection, providing near constant opportunity for recombination (*Tandel et al., 2019*). This is reminiscent of the importance of chromosomal position and recombination for the antigenically varied VSG and var genes in *Trypanosoma brucei* and *P. falciparum*, respectively (*Dreesen et al., 2007*; *Zhang et al., 2019*). The fact that MEDLE proteins are delivered into the host cell makes them prime targets for immunity, and immune evasion may drive chromosomal organization, expression, and evolution of these genes.

# Materials and methods

**Key resources table**

| Reagent type (species) or resource | Designation | Source or reference | Identifiers | Additional information |
|---|---|---|---|---|
| Gene (*Cryptosporidium parvum*) | MEDLE2 | *Li et al., 2017* | cgd5_4590 | |
| Gene (*C. parvum*) | MEDLE1 | *Fei et al., 2018* | cgd5_4580 | |
| Gene (*C. parvum*) | MEDLE6 | This paper | cgd6_5490 | Named according to the spatial localization in the genome similarly to MEDLE2 and MEDLE1 |
| Gene (*C. parvum*) | WYLE4 | This paper | cgd8_3560 | Named for the gene family and according to the spatial localization in the genome |
| Gene (*C. parvum*) | SKSR7 | This paper | Cgd8_30 | Named for the gene family and according to the spatial localization in the genome |
| Strain, strain background (*C. parvum*) | *C. parvum* oocsts, IOWAII strain (WT) | Bunchgrass | | |
| Strain, strain background (*Mus musculus*) | *Ifng*-/-, C57BL/6J | The Jackson Laboratory | Jax 002287; RRID:IMSR_JAX:002287 | |
| Strain, strain background (*M. musculus*) | *Ddit3*tm2.1Dron | The Jackson Laboratory | Jax 005530; RRID:IMSR_JAX:005530 | |
| Strain, strain background (*M. musculus*) | C57BL/6J | The Jackson Laboratory | Jax 000664; RRID:IMSR_JAX:000664 | |
| Genetic reagent (*C. parvum*) | MEDLE2-HA | This paper | cgd5_4590 modified | Stable transgenic parasite line expressing HA |
| Genetic reagent (*C. parvum*) | WYLE4-HA | This paper | cgd8_3570 modified | Stable transgenic parasite line expressing HA |
| Genetic reagent (*C. parvum*) | SKSR7-HA | This paper | cgd8_30 modified | Stable transgenic parasite line expressing HA |

*Continued*

| Reagent type (species) or resource | Designation | Source or reference | Identifiers | Additional information |
|---|---|---|---|---|
| Genetic reagent (*C. parvum*) | MEDLE2 KO | This paper | Cgd5_4590 | Stable transgenic parasite line with one copy of MEDLE2 knocked out |
| Genetic reagent (*C. parvum*) | MEDLE1-HA | This paper | cgd5_4580 modified | Stable transgenic parasite line expressing HA |
| Genetic reagent (*C. parvum*) | MEDLE6-HA | This paper | cgd6_5490 modified | Stable transgenic parasite line expressing HA |
| Genetic reagent (*C. parvum*) | *Medle2* MEDLE1-HA | This paper | cgd5_4440 modified | Stable transgenic parasite line expressing HA |
| Genetic reagent (*C. parvum*) | MEDLE2-HA-tdNeon | This paper | cgd5_4590 modified | Stable transgenic parasite line expressing HA and tdNeon |
| Genetic reagent (*C. parvum*) | MEDLE2-mScarlet | This paper | cgd5_4590 modified | Stable transgenic parasite line expressing mScarlet |
| Genetic reagent (*C. parvum*) | MEDLE2-Bla-2A-tdTomato | This paper | cgd5_4590 modified | Stable transgenic parasite line expressing BLA and tdTomato |
| Genetic reagent (*C. parvum*) | MEDLE2-Cre | This paper | cgd5_4590 modified | Stable transgenic parasite line expressing Cre recombinase |
| Genetic reagent (*C. parvum*) | Ectopic MEDLE2-HA | This paper | cgd5_4440 modified | Stable transgenic parasite line expressing extra copy of MEDLE2 |
| Genetic reagent (*C. parvum*) | ΔSP | This paper | cgd5_4440 modified | Stable transgenic parasite line expressing extra copy of MEDLE2 (aa 21–209) |
| Genetic reagent (*C. parvum*) | KDVSLI/6A | This paper | cgd5_4440 modified | Stable transgenic parasite line expressing extra copy of MEDLE2 with KDVSLI (aa 31–36) mutated to six alanines |
| Genetic reagent (*C. parvum*) | KPVLKN/6A | This paper | cgd5_4440 modified | Stable transgenic parasite line expressing extra copy of MEDLE2 with KPVLKN (aa 73–78) mutated to six alanines |
| Genetic reagent (*C. parvum*) | KNVNLS/6A | This paper | cgd5_4440 modified | Stable transgenic parasite line expressing extra copy of MEDLE2 with KDVSLI (aa 77–82) mutated to six alanines |
| Genetic reagent (*C. parvum*) | RGLLRGLSG/9A | This paper | cgd5_4440 modified | Stable transgenic parasite line expressing extra copy of MEDLE2 with KDVSLI (aa 191–199) mutated to six alanines |
| Genetic reagent (*C. parvum*) | K31A | This paper | cgd5_4440 modified | Stable transgenic parasite line expressing extra copy of MEDLE2 with K31 mutated to alanine |
| Genetic reagent (*C. parvum*) | D32A | This paper | cgd5_4440 modified | Stable transgenic parasite line expressing extra copy of MEDLE2 with D32 mutated to alanine |
| Genetic reagent (*C. parvum*) | V33A | This paper | cgd5_4440 modified | Stable transgenic parasite line expressing extra copy of MEDLE2 with V33 mutated to alanine |
| Genetic reagent (*C. parvum*) | S34A | This paper | cgd5_4440 modified | Stable transgenic parasite line expressing extra copy of MEDLE2 with S34 mutated to alanine |
| Genetic reagent (*C. parvum*) | L35A | This paper | cgd5_4440 modified | Stable transgenic parasite line expressing extra copy of MEDLE2 with L35 mutated to alanine |

*Continued on next page*

*Continued*

| Reagent type (species) or resource | Designation | Source or reference | Identifiers | Additional information |
|---|---|---|---|---|
| Genetic reagent (*C. parvum*) | I36A | This paper | cgd5_4440 modified | Stable transgenic parasite line expressing extra copy of MEDLE2 with I36 mutated to alanine |
| Cell line (human) | HCT-8 | ATCC | CCL-224; RRID:CVCL_2478 | |
| Cell line (human) | HEK293T | ATCC | CRL-3216; RRID:CVCL_0063 | |
| Cell line (*Escherichia coli*) | GC5 | Genesee Scientific | 42-653 | Electrocompetent cells |
| Cell line (*E. coli*) | One Shot Topo10, | Invitrogen | C404003 | Electrocompetent cells |
| Transfected construct (human) | loxP GFP/RFP color switch lentivirus | GenTarget Inc | Cat#: LVP460-Neo | Transfected construct (human) |
| Biological sample (*M. musculus*) | Isolated sections of ileum | *Ifng*⁻/⁻ mice | Jax 002287; RRID:IMSR_ JAX:002287 | 6-week-old male mice |
| Antibody | Anti-HA (rat monoclonal) | MilliporeSigma | Cat# 11867431001; RRID:AB_390919 | IF (1:500), WB (1:500), IHC (1:500) |
| Antibody | Anti-Cp23 (mouse monoclonal) | LS Bio | Cat# LS-C137378; RRID:AB_10947007 | IF (1:100) |
| Antibody | Anti-alpha tubulin (mouse monoclonal) | Developmental Studies Hybridoma Bank | Cat#12G10; RRID:AB_1157911 | IF (1:1000) |
| Antibody | Anti-COWP1 (rat monoclonal) | This paper, produced by GenScript | This paper | IF (1:100) |
| Antibody | Anti-neomycin phosphotransferase II (rabbit polyclonal) | MilliporeSigma | Millipore Cat# 06-747; RRID:AB_310234 | WB (1:1000) |
| Antibody | Anti-mouse IFN gamma | Bio X Cell | Clone: XMG1.2; Cat# BE0055; RRID:AB_1107694 | In vivo 100 µg |
| Antibody | Goat anti-rat polyclonal Alexa Fluor 594 | Thermo Fisher Scientific | Cat# A-21213; RRID:AB_2535799 | IFA (1:500) |
| Antibody | Goat anti-mouse polyclonal Alexa Fluor 488 | Thermo Fisher Scientific | Cat# A-11001; RRID:AB_2534069 | IFA (1:500) |
| Strain, strain background | Alexa Fluor 647 Phalloidin | Thermo Fisher Scientific | A22287; RRID:AB_2620155 | IFA (1:1000) |
| Antibody | IRDye 800CW goat anti-rat IgG | LI-COR | 926-32219; RRID:AB_1850025 | WB (1:10,000) |
| Antibody | IRDye 680RD goat anti-rabbit IgG | LI-COR | 926-68071; RRID:AB_2721181 | WB (1:10,000) |
| Recombinant DNA reagent | Cas9 cgd5_4590 (plasmid) | This paper | | Guide targeting C terminus of MEDLE2 |
| Recombinant DNA reagent | Cas9 cgd8_3560 (plasmid) | This paper | | Guide targeting C terminus of WYLE4 |
| Recombinant DNA reagent | Cas9 cgd8_30 (plasmid) | This paper | | Guide targeting C terminus of SKSR7 |
| Recombinant DNA reagent | Cas9 cgd5_4580 (plasmid) | This paper | | Guide targeting C terminus of MEDLE1 |

*Continued on next page*

*Continued*

| Reagent type (species) or resource | Designation | Source or reference | Identifiers | Additional information |
|---|---|---|---|---|
| Recombinant DNA reagent | Cas9 Cgd6_5490 (plasmid) | This paper | | Guide targeting C terminus of MEDLE6 |
| Recombinant DNA reagent | Cas9 Tk guide int (plasmid) | *Tandel et al., 2019* | | Guide targeting internal cgd5_4440 |
| Recombinant DNA reagent | Cas9 MEDLE2 KO (plasmid) | This paper | | Guide targeting internal MEDLE2 |
| Recombinant DNA reagent | Lic HA (plasmid) | This paper | | *Crypto* expression vector for HA tagging |
| Recombinant DNA reagent | Lic tdTomato KO (plasmid) | This paper | | *Crypto* expression vector for replacing gene KO with tdTomato |
| Recombinant DNA reagent | Lic *medle2*-MEDLE1 HA (plasmid) | This paper | | *Crypto* expression vector with medle2 promoter driving MEDLE1-HA expression |
| Recombinant DNA reagent | Lic HA-2A-TdNeon (plasmid) | This paper | | *Crypto* expression vector for HA tagging and cytoplasmic tdNeon |
| Recombinant DNA reagent | Lic mScarlet (plasmid) | This paper | | *Crypto* expression vector for mScarlet tagging |
| Recombinant DNA reagent | Lic Bla-2A-TdTomato (plasmid) | This paper | | *Crypto* expression vector for BLA tagging and cytoplasmic tdTomato |
| Recombinant DNA reagent | Lic Cre (plasmid) | This paper | | *Crypto* expression vector for Cre tagging |
| Recombinant DNA reagent | Lic Extra MEDLE2-HA (plasmid) | This paper | | *Crypto* expression vector for extra copy of MEDLE2-HA (MEDLE2 promoter) |
| Recombinant DNA reagent | Lic ΔSP MEDLE2-HA (plasmid) | This paper | | *Crypto* expression vector for extra copy of MEDLE2-HA (aa 21–209) |
| Recombinant DNA reagent | Lic KDVSLI/6A-HA (plasmid) | This paper | | *Crypto* expression vector for extra copy of MEDLE2-HA with KDVSLI (aa 31–36) mutated to six alanines |
| Recombinant DNA reagent | Lic KPVLKN/6A-HA (plasmid) | This paper | | *Crypto* expression vector for extra copy of MEDLE2-HA with KPVLKN (aa 73–78) mutated to six alanines |
| Recombinant DNA reagent | Lic KNVNLS/6A-HA (plasmid) | This paper | | *Crypto* expression vector for extra copy of MEDLE2-HA with KNVNLS (aa 77–82) mutated to six alanines |
| Recombinant DNA reagent | Lic RGLLRGLSG/9A-HA (plasmid) | This paper | | *Crypto* expression vector for extra copy of MEDLE2-HA with RGLLRGLS (aa 191–199) mutated to six alanines |
| Recombinant DNA reagent | Lic K31A-HA (plasmid) | This paper | | *Crypto* expression vector for extra copy of MEDLE2-HA with K31 mutated to alanine |
| Recombinant DNA reagent | Lic D32A-HA (plasmid) | This paper | | *Crypto* expression vector for extra copy of MEDLE2-HA with D32 mutated to alanine |
| Recombinant DNA reagent | Lic V33A-HA (plasmid) | This paper | | *Crypto* expression vector for extra copy of MEDLE2-HA with V33 mutated to alanine |
| Recombinant DNA reagent | Lic S34A-HA (plasmid) | This paper | | *Crypto* expression vector for extra copy of MEDLE2-HA with S34 mutated to alanine |

*Continued on next page*

*Continued*

| Reagent type (species) or resource | Designation | Source or reference | Identifiers | Additional information |
|---|---|---|---|---|
| Recombinant DNA reagent | Lic L35A-HA (plasmid) | This paper | | *Crypto* expression vector for extra of MEDLE2-HA with L35 mutated to alanine |
| Recombinant DNA reagent | Lic I36A-HA (plasmid) | This paper | | *Crypto* expression vector for extra copy of MEDLE2-HA with I36 mutated to alanine |
| Recombinant DNA reagent | mEGFP-Lifeact-7 (plasmid) | Addgene | # 54610 | Used as a mammalian expression vector to clone codon optimized MEDLE2 into |
| Recombinant DNA reagent | GFP-only | This paper | | Removed Lifeact domain from Addgene plasmid #54610 for a GFP-only control plasmid |
| Recombinant DNA reagent | Recod MEDLE2-GFP (plasmid) | This paper | | Human codon optimized MEDLE2 (aa 21–209) with a GFP tag |
| Recombinant DNA reagent | GRA16-GFP | This paper | | *T. gondii* GRA16 (aa 24–505) with GFP tag. |
| Recombinant DNA reagent | Recod MEDLE2-HA (plasmid) | This paper | | Human codon optimized MEDLE2 (aa 21–209) with a HA tag. |
| Recombinant DNA reagent | Recod KDVSLI/6A-HA (plasmid) | This paper | | Human codon optimized MEDLE2 (aa 21–209) with a HA tag and KDVSLI (aa 31–36) mutated to six alanines |
| Recombinant DNA reagent | Recod KPVLKN/6A-HA (plasmid) | This paper | | Human codon optimized MEDLE2 (aa 21–209) with a HA tag and KPVLKN (aa 73–78) mutated to six alanines |
| Recombinant DNA reagent | Recod KNVNLS/6A-HA (plasmid) | This paper | | Human codon optimized MEDLE2 (aa 21–209) with a HA tag and KNVNLS (aa 77–82) mutated to six alanines |
| Recombinant DNA reagent | Recod RGLLRGLSG/9A-HA (plasmid) | This paper | | Human codon optimized MEDLE2 (aa 21–209) with a HA tag and RGLLRGLS (aa 191–199) mutated to six alanines |
| Sequence-based reagent | Recodonized MEDLE2 | Integrated DNA Technologies | | MEDLE2 (aa 21–209) codon optimized for human expression See *Supplementary file 1* for sequence |
| Sequence-based reagent | PCR primers | This paper | | Please see *Supplementary file 1* |
| Commercial assay or kit | DNeasy Blood & Tissue Kit | QIAGEN | Cat# 69504 | |
| Commercial assay or kit | ZymoPureII Plasmid Maxiprep Kit | Zymo Research | Cat# 11-555B | |
| Commercial assay or kit | Nano-Glo Luciferase Assay System | Promega | Cat# N1130 | |
| Commercial assay or kit | SF Cell Line 4D X Kit L | Lonza | Cat# V4XC-2024 | |
| Commercial assay or kit | LiveBLAzer FRET-B/G Loading Kit | Thermo Fisher | Cat# K1095 | |
| Commercial assay or kit | ZeroBlunt TopoTA Kit | Invitrogen | Cat# 450245 | |
| Commercial assay or kit | RNeasy Microkit | QIAGEN | Cat# 74004 | |
| Commercial assay or kit | SMART cDNA synthesis kit | Takara | Cat# 635040 | |

*Continued on next page*

*Continued*

| Reagent type (species) or resource | Designation | Source or reference | Identifiers | Additional information |
|---|---|---|---|---|
| Commercial assay or kit | Nextera XT DNA Library Prep Kit | Illumina | Cat# FC-131-1096 | |
| Commercial assay or kit | RNeasy MiniKit | QIAGEN | Cat# 74104 | |
| Commercial assay or kit | QIAshredder | QIAGEN | Cat# 79656 | |
| Commercial assay or kit | SuperScript First Strand Synthesis kit | Thermo Fisher | Cat#18091050 | |
| Commercial assay or kit | Lipofectamine 3000 | Thermo Fisher | Cat# L3000015 | |
| Chemical compound, drug | Paromomycin | Gemini | Cat# 400-155P | Used 16 g/L water |
| Chemical compound, drug | Brefeldin A (BFA) | BioLegend | Cat# 420601 | Used 10 µg/mL |
| Chemical compound, drug | Thapsigargin | MedChemExpress | HY-13433 | Used 1 µM |
| Chemical compound, drug | GSK2606414 | MedChemExpress | HY-18072 | Used 30 nm |
| Chemical compound, drug | KIRA6 | MedChemExpress | HY-19708 | Used 500 nm |
| Software, algorithm | Prism 8 | GraphPad | RRID:SCR_002798 | |
| Software, algorithm | ImageJ | Fiji | RRID:SCR_003070 | |
| Software, algorithm | FlowJo v10, LLC | TreeStar | RRID:SCR_008520 | |
| Software, algorithm | Kallisto v0.44.0 | BioConductor (*Bray et al., 2016*) | Pachter Lab | |
| Software, algorithm | Limma-Voom | BioConductor (*Law et al., 2014*; *Ritchie et al., 2015*) | | |
| Software, algorithm | Bioconductor tximport | BioConductor (*Robinson et al., 2010*) | DOI: 10.18129/B9.bioc.tximport | |
| Software, algorithm | Molecular Signatures Database (MSigDB) | UC San Diego and Broad Institute (*Mootha et al., 2003*; *Subramanian et al., 2005*) | https://www.gsea-msigdb.org/gsea/msigdb | |
| Software, algorithm | CryptoDB | VEuPathDB | cryptodb.org | |
| Other | Fluorescin *Vicia villosa* lectin stain | Vector Labs | Cat# FL-1231-2 | IF (1:1000) |
| Other | DAPI stain | Invitrogen | Cat# D1306 | Flow cytometry (1 µg/mL) |
| Other | Hoechst 33342 | Thermo Fisher | Cat# H3570 | IF (1:10,000) |
| Other | Alexa Fluor 647 Phalloidin | Thermo Fisher | Cat# A22287 | IF (1:1000) |

## Contact for reagent and resource sharing

For access to reagents or parasite strains used in this study, please contact Dr. Boris Striepen: Tel.: 1-215-573-9167; fax: 1-215-746-2295; e-mail: striepen@upenn.edu.

## Mouse models of infection

*Ifng*[-/-] (stock no: 002287; RRID:IMSR_JAX:002287) were purchased from Jackson Laboratory and maintained as a breeding colony at the University of Pennsylvania. All mice used in this study ranged from 4 to 12 weeks of age. We note that both male and female *Ifng*[-/-] mice were used to generate and propagate *C. parvum* transgenic parasite lines, without a difference being noted in parasite shedding. *Ddit3-/-* (stock no: 005530) and C57BL/6J (stock no: 000664) mice were purchased from Jackson Laboratory. All protocols for animal care were approved by the Institutional Animal Care and Use Committee of the University of Georgia (protocol A2016 01-028-Y1-A4) and the Institutional Animal Care and Use Committee of the University of Pennsylvania (protocol #806292).

## Cell lines

HCT-8 (ATCC: CCL-224TM; RRID:CVCL_2478) and HEK293T (ATCC: CRL-3216TM; RRID:CVCL_0063) cell lines were purchased from ATCC, mycoplasma tested, and used for all experiments once confirmed to be mycoplasma negative.

## Parasite strains

*C. parvum* transgenic strains were made and propagated in *Ifng*-/- mice (stock no: 002287; RRID:IMSR_JAX:002287). Oocysts were then purified from fecal collections using sucrose flotation followed by a cesium chloride gradient (see Materials and methods). All *C. parvum* oocysts used in this study as WT controls, as well as to generate transgenic strains, are on the IOWAII strain background, purchased from Bunchgrass Farms (Dreary, ID).

## Plasmid construction

Guide oligonucleotides (Sigma-Aldrich, St. Louis, MO) were introduced into the *C. parvum* Cas9/U6 plasmid by restriction cloning, as detailed in *Pawlowic et al., 2017*. All plasmids encoding epitope tags, as well as for ectopic MEDLE2 expression, were constructed by Gibson assembly using NEB Gibson Assembly Master Mix (New England Biolabs, Ipswich, MA). A linear repair template was generated by PCR. See *Supplementary file 1* for a complete list of primers used for this study.

## Generation of transgenic parasites

Transgenic parasites were derived as previously described (*Sateriale et al., 2020*). Briefly, $5 \times 10^7$ *C. parvum* oocysts were bleached on ice, washed in 1× PBS, and incubated in sodium taurodeoxycholate. Excysted sporozoites were resuspended in transfection buffer supplemented with a total of 100 μg DNA (comprising 50 μg of Cas9/gRNA plasmid and 50 μg of repair template generated by PCR) and nucleofected using an Amaxa 4D nucleofector (Lonza, Basel, Switzerland). Transfected parasites were resuspended in PBS and administered to *Ifng*-/- mice. Mice were pretreated with antibiotics for 1 week preceding infection and with sodium bicarbonate immediately before parasite administration (*Sateriale et al., 2020*). Mice received 16 mg/mL paromomycin in drinking water for selection. Transgenic parasites were detected by measuring fecal nanoluciferase activity and purified from feces using sucrose flotation followed by a cesium chloride gradient and stored in PBS at 4°C (*Sateriale et al., 2020*).

## Nanoluciferase assay to monitor parasite shedding

20 mg of fecal material was dissolved in nanoluciferase lysis buffer and mixed 1:1 with nanoluciferase substrate/nanoluciferase Assay Buffer (1:50) in a white-bottom plate. Relative luminescence was read using a Promega GloMax Plate Reader.

## Integration PCR to confirm generation of transgenic parasites

DNA was purified from excysted sporozoites using the QIAGEN DNeasy Blood and Tissue kit (QIAGEN 69504). PCR primers were designed to anneal outside of the 5′ and 3′ homology arms used to direct homologous recombination and matched with primers annealing to the nanoluciferase reporter gene or the neomycin selection marker, respectively. Primers for the thymidine kinase gene served as control, unless otherwise noted. Where indicated, amplicons were cloned using the ZeroBlunt TopoTA kit (Invitrogen 450245) and transformed into One Shot Topo10 Chemically Competent *Escherichia coli* (Invitrogen C404003). Individual colonies were miniprepped and sequenced.

## In vitro infection and immunofluorescence assay

Coverslips seeded with human ileocecal adenocarcinoma cells (HCT-8) (ATCC CCL-244; RRID:CVCL_2478) were infected when 80% confluent with 200,000 purified oocysts (bleached, washed, and resuspended in RPMI medium containing 1% serum). For time-course infections, parasites were allowed to invade for 3 hr, then medium was removed, and the cells were washed with PBS to remove unexcysted oocysts and replaced with fresh RPMI medium with 1% serum. At indicated time points, cells were washed with PBS, and successively fixed and permeabilized with PBS supplemented with 4% paraformaldehyde or 0.1% Triton X-100 for 10 min each (Sigma). Coverslips were blocked with 1% bovine serum albumin (BSA) (Sigma). Antibodies were diluted in blocking solution.

The was rat monoclonal anti-HA (MilliporeSigma, Burlington, MA; RRID:AB_390919) and was used as primary antibody (1:500) and goat anti-rat polyclonal Alexa Fluor 594 (Thermo Fisher, Waltham, MA; RRID:AB_2535799) as secondary along with *V. villosa* lectin (Vector Labs Burlingame, CA). Host and parasite nuclei were stained with Hoechst 33342 (Thermo Fisher). Slides were imaged using a Zeiss LSM710 Confocal microscope or a Leica Widefield microscope. Experimental slides were prepared and imaged in duplicate for a minimum of two biological replicates.

## Immunohistochemistry on infected intestine

Infected *Ifng*<sup>-/-</sup> mice (RRID:IMSR_JAX:002287) were euthanized at day 12 during peak infection, and the distal 1/3 of the small intestine was dissected. The tissue was washed with PBS and 'swiss-rolled' and fixed overnight in 4% paraformaldehyde at 4°C, placed in 30% sucrose in PBS for cryoprotection, and mounted with OCT compound (Tissue-Tek, Sakura Finetek, Japan) and frozen. Cryomicrotome sections were permeabilized and blocked and labeled as described above (Alexa Fluor 647 Phalloidin [RRID:AB_2620155; Thermo Fisher] was used in addition). Sections were prepared in duplicate and imaged using a Zeiss LSM710 Confocal Microscope.

## Poly-L-lysine treatment of coverslips and sporozoite IFA

Sterile coverslips were treated with poly-L-lysine (Sigma), washed with water for 5 min, and airdried. Sporozoites suspended in PBS were allowed to settle on treated coverslip for 1 hr prior to fixing and IFA. Primary antibodies used were mouse anti-Cp23 (1:100) (LS Bio Seattle, WA; RRID:AB_10947007) and rat monoclonal anti-HA (RRID:AB_390919) (1:500). Secondary antibodies include goat anti-rat polyclonal Alexa Fluor 594 (RRID:AB_2535799)and goat anti-mouse polyclonal Alexa Fluor 488 (RRID:AB_2534069) (both 1:1000, Thermo Fisher).

## BFA treatment during *C. parvum* infection

Excysted oocysts were allowed to invade HCT-8 coverslip cultures (RRID:CVCL_2478) for 3 hr, unexcysted oocysts were removed by PBS wash, and cultures were replaced with medium supplemented with 1% serum and 10 µg/mL BFA from a 1000× stock in DMSO. Medium supplemented with carrier alone served as control. Cultures were fixed and processed 10 hr post infection.

## Live imaging of beta-lactamase reporter assay

$1 \times 10^6$ WT and MEDLE2-BLA oocysts were used to infect HCT-8 cells (RRID:CVCL_2478) in a 35 mm glass-bottom dish (MatTek Life Sciences, Ashland, MA). After 24 hr, the medium was replaced with RPMI medium containing CCF4-AM substrate from the LiveBLAzer FRET-B/G Loading Kit (Thermo Fisher Scientific). Cells were incubated in the dark at 37°C for 1 hr, washed with PBS three times, and live imaged using a Leica SP5 Confocal Microscope using a water immersion lens.

## Cre recombinase reporter assay by flow cytometry

Pre-made lentivirus was used to transform HCT-8 cells (RRID:CVCL_2478) with a loxP GFP/RGP color switch cassette (GenTarget Inc, San Diego, CA). Cells were selected with 400 mg/mL neomycin (MilliporeSigma) for 14 days and validated by transfection with 5 µg Cre recombinase plasmid using Lipofectamine P3000 (Thermo Fisher Scientific). After 24 hr and 48 hr, cells were trypsinized and flow sorted using a LSRFortessa (BD Biosciences, San Jose, CA) and data were analyzed with FlowJo v10 software (TreeStar) (RRID: SCR_008520).

$1 \times 10^6$ WT and MEDLE2-Cre oocysts were used to infect six-well cultures of Lox GFP/RFP color switch cells. After 48 hr, cells were trypsinized and resuspended in 1 mL PBS. 300 µL were used for nanoluciferase assay and 700 µL cells for flow cytometry. Forward and side scatter was used to gate viability, untransfected uninfected cells to establish the green gate, and Cre recombinase transfected cells for the red gate (three biological replicates for each condition).

## Western blot on *C. parvum* infected cells

HCT-8 cultures (RRID:CVCL_2478) infected with $5 \times 10^6$ oocysts for 48 hr were treated with Trypsin 0.25% EDTA (Thermo Fisher Scientific), pelleted, and flash frozen in liquid nitrogen. Cell pellets were lysed in Pierce IP Lysis Buffer (Thermo Fisher Scientific), supplemented 1:100 with both protease inhibitor cocktail (Sigma) and benzonase nuclease (MilliporeSigma). Lysates were incubated on ice for

15 min, sonicated (80% amplitude, 10 s pulses, rest on ice for 1 min between three times), cleared by centrifugation (20,000 × *g*, 10 min, 4°C), mixed with freshly prepared Laemmli Sample buffer (Milli-poreSigma) + β-mercaptoethanaol (1:20) (Sigma), boiled and loaded on a 12% Mini-PROTEAN TGX Precast Protein Gel (Bio-Rad, Hercules, CA) run at 70 V for 2.5 hr. Gels were transferred to 0.45 µm pore size nitrocellulose membrane (Thermo Fisher Scientific) overnight at 0.02 A at 4°C. The membrane was blocked for 1 hr with Intercept (TBS) Protein-Free Blocking Buffer (LI-COR, Lincoln, NE), antibodies we diluted in blocking solution with 0.01% Tween20 (Sigma) using rat monoclonal anti-HA 1:500 (MilliporeSigma; RRID:AB_390919), and rabbit anti-neomycin phosphotransferase II 1:1000 (Milli-poreSigma; RRID:AB_310234) as primary and IRDye 800CW goat anti-rat IgG (RRID:AB_1850025) and IRDye 680RD goat anti-rabbit IgG (RRID:AB_2721181) (both 1:10,000, LI-COR) as secondary antibody. Washed membranes were imaged using an Odyssey Infrared Imaging System v3.0 (LI-COR).

## Generation of MEDLE2 mutant plasmids for host cell transfection

Human codon optimized MEDLE2 lacking the N-terminal signal peptide (aa 21–209) was synthesized by Integrated DNA Technologies (IDT, Coralville, IA) and cloned into the mEGFP-Lifeact-7 mammalian expression plasmid (Addgene #54610), replacing Lifeact-GFP and appending a 3× HA tag. Point mutations were engineered by Gibson cloning. HEK293T cells (ATCCCRL-3216; RRID:CVCL_0063) were transfected with 5 µg of each plasmid using Lipofectamine P3000 (Thermo Fisher Scientific). 24 hr post transfection, cells were harvested and processed for western blot analysis. Additionally, a MEDLE2-EGFP plasmid was cloned by Gibson cloning to introduce human codon optimized MEDLE2 lacking the N-terminal signal peptide (aa 21–209) into the same mEGFP-Lifeact-7 mammalian expression plasmid, replacing Lifeact. A GFP-only-expressing plasmid was engineered removing the Lifeact from Addgene plasmid #54610. Similarly, *T. gondii* GRA16 omitting the sequence encoding the N-terminal signal peptide (aa 24–505) was amplified from gDNA of *T. gondii* strain ME49 parasites and cloned into the mEGFP-Lifeact-7 mammalian expression plasmid in place of Lifeact.

## Flow cytometry analysis of transfected cells

HEK293T cells (RRID:CVCL_0063) were subjected to lipofection with 25 µg GFP-only plasmid or MEDLE2-GFP plasmid, grown for 24 hr, trypsinized, washed, and resuspended in PBS with DAPI and passed through a 40 µM filter (BD Biosciences). Cell viability was gated based upon DAPI staining. Untransfected HEK293T served as negative control and GFP-expressing HEK293T cells as positive control to establish gates. 10,000 green, single cells were double sorted using an Aria C flow cytometer first into PBS then into lysis buffer (three biological replicates for each condition).

## RNA extraction sequencing and data analysis

Total RNA was extracted using the QIAGEN RNeasy Microkit (QIAGEN, Germantown, MD) and input RNA was quality controlled and quantified using a Tape Station 4200 (Agilent Technologies, Santa Clara, CA). cDNA synthesis was performed following the clonTechSMART-seq cDNA synthesis protocol (15 cycles). Following DNA cleanup, a Nextera library was prepared and nucleic acid was quantified using the Qubit 3 Fluorometer (Thermo Fisher Scientific). Samples were pooled for RNA-sequencing of 4 nM of total cDNA, and sequencing was performed using a NextSeq 500 Instrument (Illumina Inc, San Diego, CA).

RNAseq reads were pseudo-aligned to the Ensembl *Homo sapiens* reference transcriptome v86 using kallisto v0.44.0 (*Bray et al., 2016*). In R, transcripts were collapsed to genes using Bioconductor tximport (*Robinson et al., 2010*) and differentially expressed genes were identified using Limma-Voom (*Law et al., 2014*; *Ritchie et al., 2015*). The MEDLE2 transcription response dataset can be found under GEO accession number GSE174117. GSEA was performed using the GSEA software and the annotated gene sets of the Molecular Signatures Database (MSigDB) (*Mootha et al., 2003*; *Subramanian et al., 2005*). The MEDLE2 signature was generated from the differentially expressed genes and read into GSEA to evaluate its presence in published datasets of *C. parvum* infection (*Niko-laev et al., 2020*; *Saxena et al., 2017*).

## qPCR for MEDLE2 response genes from infected mice

8-week-old *Ifng*[-/-] mice (RRID:IMSR_JAX:002287) were infected with 10,000 *C. parvum* oocysts, and the infection was tracked by fecal nanoluciferase activity. Infected mice (n = 3) and uninfected controls

(n = 3) were euthanized after 10 days, and the distal 1/3 of the small intestine was removed. The tissue was washed with 1× PBS until clear of fecal material and then cut longitudinally. 5 mm diameter gut punches were made and preserved in RNA*later* solution (Thermo Fisher Scientific). RNA was extracted from tissue samples using the QIAGEN RNeasy MiniKit (QIAGEN) following homogenization with a bead beater and passage through a QIAshredder (QIAGEN). 5 µg of cDNA was reverse transcribed using the SuperScript First Strand Synthesis kit following the manufacturer's instructions for use with OligoDT (Thermo Fisher Scientific). qPCR was performed using a Viia7 Real-time PCR System (Thermo Fisher Scientific), and relative gene expression was determined using the ΔΔCT method.

## qPCR for MEDLE2 response genes in transfected cells

5 µg plasmid (GFP-only, MEDLE2-GFP, GRA16-GFP) was introduced to HEK293T cells (RRID:CVCL_0063) by Lipofectamine transfection. Cells were grown for 24 hr and then the medium was removed and replaced with RLT lysis buffer from the QIAGEN RNeasy MiniKit (QIAGEN) and passaged through a QIAshredder (QIAGEN) to begin the RNA extraction protocol. Untransfected HEK293T served as negative control. 5 µg of cDNA was reverse transcribed using the SuperScript First Strand Synthesis kit, and qPCR was performed using the methods described above with primers specific for GAPDH (control), GFP, CHAC1, DDIT3, NUPR1, and TRIB3.

## In vitro growth assay in the presence of UPR inhibitors

HCT-8 cells (RRID:CVCL_2478) were grown to 80% confluency in 96-well plates. 2 hr prior to infection, the medium was removed from the plate and replaced with RPMI medium containing DMSO (vehicle), 1 µM thapsigargin, 30 nM GSK2606414, or 500 nM KIRA6, all from MedChemExpress (Monmouth Junction, NJ). Nanoluciferase expressing *C. parvum* parasites were excysted, and 10,000 parasites were used to infect each well. After 24 hr, medium was removed from the wells and replaced with nanoluciferase lysis buffer and incubated for 5 min. The lysate was then mixed 1:1 with nanoluciferase substrate/nanoluciferase Assay Buffer (1:50) in a white-bottom plate and relative luminescence read using a Promega GloMax Plate Reader.

## Infection of *DDIT3* KO mice

6–9-week-old *Ddit3-/-* (stock no: 005530; RRID:IMSR_JAX:005530) and C57BL/6J (stock no: 000664; RRID:IMSR_JAX:000664) mice were purchased from Jackson Laboratory. Mice were treated with 100 mg of InVivo Mab anti mouse-IFN gamma antibody Bio X Cell (Lebanon, NH; RRID:AB_1107694) 1 day prior to infection and again at day 2 of infection. Mice were infected with 10,000 MEDLE2-HA-tdNeon oocysts, and feces were collected every 2 days to measure fecal luminescence by nanoluciferase activity as described above.

## Quantification and statistical methods

GraphPad Prism (RRID:SCR_002798) was used for all statistical analyses. When measuring the difference between two populations, a standard *t*-test was used. For datasets with three or more experimental groups, a one-way ANOVA with Dunnett's multiple comparison test was used. Simple linear regression was used to determine the goodness-of-fit curve for the number of MEDLE2-expressing cells and intracellular parasites. Quantification of imaging experiments was performed using ImageJ (RRID:SCR_003070) macros programmed to count both parasites and host cell nuclei in blinded images that were captured using a scanning function to avoid bias during acquisition.

## Acknowledgements

This work was supported in part by funding from the National Institutes of Health through grants to BS (R01AI127798 and R01AI112427), and fellowships and career awards to JED (T32AI007532), AS (K99AI137442), and JAG (T32A1055400). We thank Andrea Stout, Jasmine Zhao (Cell and Developmental Biology Microscopy Core), Gordon Ruthel (Penn Vet Imaging Core), Dan Beiting (Center for Host Microbial Interactions), and Patty Costello (Comparative Pathology Core) for help with microscopy, RNA sequencing, and histology, respectively, and Jacek Gaertig and Phillip Scott for sharing reagents.

## Additional information

### Funding

| Funder | Grant reference number | Author |
| --- | --- | --- |
| National Institute of Allergy and Infectious Diseases | R01AI127798 | Boris Striepen |
| National Institute of Allergy and Infectious Diseases | R01AI112427 | Boris Striepen |
| National Institute of Allergy and Infectious Diseases | T32AI007532 | Jennifer E Dumaine |
| National Institute of Allergy and Infectious Diseases | K99AI137442 | Adam Sateriale |
| National Institute of Allergy and Infectious Diseases | T32A1055400 | Jodi A Gullicksrud |

The funders had no role in study design, data collection and interpretation, or the decision to submit the work for publication.

### Author contributions

Jennifer E Dumaine, Conceptualization, Formal analysis, Funding acquisition, Investigation, Methodology, Writing - original draft, Writing – review and editing; Adam Sateriale, Conceptualization, Funding acquisition, Methodology, Supervision, Writing – review and editing; Alexis R Gibson, Formal analysis, Investigation, Methodology, Writing – review and editing; Amita G Reddy, Emma N Hunter, Investigation, Writing – review and editing; Jodi A Gullicksrud, Joseph T Clark, Investigation, Methodology, Writing – review and editing; Boris Striepen, Conceptualization, Funding acquisition, Project administration, Supervision, Writing - original draft, Writing – review and editing

### Author ORCIDs

Jennifer E Dumaine ![ORCID] http://orcid.org/0000-0002-5975-4523
Alexis R Gibson ![ORCID] http://orcid.org/0000-0003-1078-4841
Boris Striepen ![ORCID] http://orcid.org/0000-0002-7426-432X

### Ethics

All animals used in this study were handled and cared for in accordance with approved Institutional Animal Care and Use Committee protocols at the University of Georgia (protocol A2016 01-028-Y1-A4) and the University of Pennsylvania (protocol #806292).

### Decision letter and Author response

Decision letter https://doi.org/10.7554/eLife.70451.sa1
Author response https://doi.org/10.7554/eLife.70451.sa2

---

# Additional files

### Supplementary files
• Transparent reporting form
• Supplementary file 1. Primer sequences used for this study.
• Source code 1. Supplemental code detailing the R packages used for analysis of the MEDLE2 transfection RNAsequencing dataset.

### Data availability
The RNA sequencing dataset generated from the MEDLE2 transfection experiment has been deposited in GEO under accession number GSE174117. Source code and data files for this dataset were provided. Furthermore, numerical source data used for imaging quantification experiments in Figures 2 and 3 were provided.

The following dataset was generated:

| Author(s) | Year | Dataset title | Dataset URL | Database and Identifier |
|---|---|---|---|---|
| Dumaine JE, Sateriale A, Gibson AR, Reddy AG, Gullicksrud JA, Hunter EN, Clark JT, Striepen B | 2021 | The enteric pathogen Cryptosporidium parvum exports proteins into the cytoplasm of the infected host cell | https://www.ncbi.nlm.nih.gov/geo/query/acc.cgi?acc=GSE174117 | NCBI Gene Expression Omnibus, GSE174117 |

The following previously published datasets were used:

| Author(s) | Year | Dataset title | Dataset URL | Database and Identifier |
|---|---|---|---|---|
| Nikolaev M, Mitrofanova O, Broguiere N, Geraldo S, Dutta D, Tabata Y, Elci B, Brandenberg N, Kolotuev I, Gjorevski N, Clevers H, Lutolf MP | 2020 | Homeostatic mini-intestines through scaffold-guided organoid morphogenesis | https://www.ncbi.nlm.nih.gov/geo/query/acc.cgi?acc=GSE148366 | NCBI Gene Expression Omnibus, GSE148366 |
| Saxena K, Simon LM, Zeng XL, Blutt SE, Crawford SE, Sastri NP, Karandikar UC, Ajami NJ, Zachos NC, Kovbasnjuk O, Donowitz M, Conner ME, Shaw CA, Estes MK | 2017 | RNA-sequencing of human intestinal enteroids infected with or without human rotavirus (strain Ito) | https://www.ncbi.nlm.nih.gov/geo/query/acc.cgi?acc=GSE90796 | NCBI Gene Expression Omnibus, GSE90796 |

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
