## [Editor Report]

Apicomplexa parasites export proteins into their infected cells to modulate/co-opt signaling pathways for their intracellular development. This study demonstrates for the first time the export of a protein, MEDLE2, from *Cryptosporidium parvum*, and characterizes its targeting signal as well as its effector function in the infected host cell.

---

## [Decision Letter]

**Decision letter after peer review:**

Thank you for submitting your article "The enteric pathogen *Cryptosporidium parvum* exports proteins into the cytoplasm of the infected cell" for consideration by *eLife*. Your article has been reviewed by 3 peer reviewers, one of whom is a member of our Board of Reviewing Editors, and the evaluation has been overseen by Dominique Soldati-Favre as the Senior Editor. The following individual involved in review of your submission has agreed to reveal their identity: Daniel E Goldberg (Reviewer #3).

Essential revisions:

All three reviewers agree that this is an interesting and important study, and that the work is elegant and clearly presented. However they also raised a number of points, as detailed below in the recommendations to the authors, which have to be addressed to strengthen the manuscript. Specifically, the following key points need your full attention:

1) A key question is whether the observed ER stress response is a byproduct of infection or is triggered by the parasite to promote its growth and/or survival. Some key additional experiments are required to address this issue (for example transfection of GRA protein, testing the effects of UPR inhibitors on infection).

2) The host cell transcriptome data should be presented in more details, including a list of differentially expressed genes, and a more thorough discussion of which UPR pathways are affected.

Importantly beside the Essential Revisions, please consider the recommendations made by the individual reviewers and address all the specific points as far as possible in the revision.

*Reviewer #1:*

This is an interesting study reporting that the intracellular parasite Cryptosporidium, an understudied yet medically relevant apicomplexan pathogen, exports proteins into the cytoplasm of infected cells. The manuscript is very well written and describes a large amount of work based on state of the art genetic approaches.

The authors used genetic tagging to investigate the cellular localization of members of multigene families, and identified MEDLE2 as a protein that is efficiently exported into the cytoplasm of infected cells, both in cultured cells and infected mice. The authors found that MEDLE2 export occurs after invasion, indicative of a translocon-mediated process, and is blocked when the protein is fused to ordered reporters. Using a series of mutants, the authors identify a N-terminal host targeting motif and a critical Leucine residue required for export of MEDLE2. They also document MEDLE2 protein cleavage during export. Finally, RNAseq analysis of cells transfected with MEDLE2 revealed a subset of differentially expressed host genes, pointing at a possible ER stress response of the host cell induced by expression of MEDLE2. These data were confirmed on a small subset of genes using qPCR in infected mice.

While the concept of protein export has been extensively studied in other apicomplexan parasites, this study offers additional insights into host-parasite interactions and a counterpoint to protein translocation mechanisms in Plasmodium and Toxoplasma. MEDLE2 export shares some aspects with protein export in these parasites, but also points at possible divergence, as the translocon elements present in Plasmodium spp or *T. gondii* are not found in Cryptosporidium. Although the export machinery remains to be identified, this study opens perspectives to study protein export as a target for therapeutic strategies against cryptosporidiosis. Initial characterization of the host cell response revealed a transcriptional signature of ER stress and unfolded protein response, but this aspect of the manuscript is rather superficial.

In summary, this well conducted study provides a comprehensive characterization of protein export in Cryptosporidium-infected cells. This work will stimulate future studies to identify the export machinery. A key question at this stage is the function of MEDLE2 and other exported proteins, and whether the host ER stress response is a bystander effect or a specific response induced by the parasite to promote its growth or survival.

The host cell response could be explored in more details. There are several UPR pathways and it is not clear whether canonical UPR proteins are affected by MEDLE2. A previous study showed that *C. parvum* induces an ER response in HCT-8 cells (PMID 23986438), and this paper should be discussed here.

The RNAseq data (Figure 6D) shows that only a small fraction of ER stress response genes are significantly upregulated, and there are other genes that are even more upregulated. What are these genes? A full list of DGE should be provided as a supplemental table and discussed in the text.

A key question is whether the observed ER stress response is a byproduct of infection or is triggered by the parasite to promote its growth and/or survival. There are a number of simple experiments, based on knockout or knockdown of key UPR components, or using agonists or chemical inhibitors, which could provide elements of response.

Another open question is the function of MEDLE2. Ectopic expression via transfection of MEDLE2 in host cells may not reflect the natural responses during infection. While the presence of multiple copies of MEDLE genes is probably a major obstacle for genetic studies, the authors could test whether the host cell response differs when MEDLE2 is exported (wildtype protein) or not (mutant versions). This would provide stronger evidence that in the context of infection MEDLE2 is directly responsible (at least in part) for the observed host cell response. Related to this point, line 28 and 75: there is no evidence that MEDLE2 is a pathogenesis factor

MEDLE2 processing shares common features with processing in Plasmodium/Toxoplasma, where aspartyl proteases are involved. Did the authors try to block MEDLE2 export with aspartyl protease inhibitors?

The image quality of Fig5D is not optimal, the MEDLE2-HA signal is very weak.

*Reviewer #2:*

This comprehensive study elegantly demonstrates that like other parasites of the phylum *Cryptosporidium parvum* exports proteins into the cytosol of infected cells by a sophisticated mechanism. The authors investigate in great detail the maturation, transit and export of the MEDLE2 protein across the *C. parvum* lifecycle in culture and infected animals. Briefly, they showed that MEDLE2 passes through the secretory pathway and holds a host targeting motif that is processed during export. MEDLE2 is a bona fide intrinsically disordered protein (IDP), i.e., the protein has no single well-defined tertiary structure under native conditions. This feature is also shared by the growing family of exported GRA proteins in *Toxoplasma gondii* and like them, MEDLE2 exports is blocked by ordered reporters.

While there is substantial evidence presented that supports the delivery of MEDLE2 in the cytosol of the infected enterocytes, the transit and secretion of other MEDLE family members is less clear. This does not detract from the discovery of a highly regulated export system of parasite-derived effectors in *C. parvum*, of which MEDLE2 is the flagship.

Based on a wide variety of approaches, the experiments were well designed with appropriate controls and biological replicates, but further clarification is needed, especially with respect to transcriptional analyses based on expression in a heterologous system of the parasite effector. MEDLE2 does not accumulate in the nucleus of infected cells, yet its ectopic expression in mammalian cells is altering the expression of genes involved in ER stress response, including changes in genes linked to the unfolded protein response. One would have to rule out the possibility that the induction of this pathway is not simply a byproduct of the transient overexpression of a disordered protein. This argument is to some extent mitigated by the demonstration that alteration of genes in this pathway has also been observed during infection with *C. parvum*, yet definitive evidence with a medle2 KO is still missing.

Overall, all the molecular and cell biological analyses and data presented are technically strong and quite convincing and support the notion of an important and novel example of a pathogen-derived modulator of the host response in *C. parvum*.

– Lines 368-371: the authors state that "It remains to be determined whether the UPR is a byproduct of host detection of the disordered protein MEDLE2 following its translocation during *C. parvum* infection, or if it is an active process deliberately triggered to promote parasite survival." A key experiment to address this dilemma rightly formulated by the authors would be to express in 293 cells in parallel to MEDLE2, instead of GFP, a mock protein with an intrinsically disordered region (e.g. a Tg exported GRA protein), and monitor by qRT-PCR the expression levels of the genes mentioned in Figure 6F.

– Ectopic expression of MEDLE1 at the TK locus (Figure 1—figure supplement 3E-F) under the control of a fragment of the MEDLE2 promoter only greatly enhances extra-vacuolar MEDLE1 secretion. How were the functional boundaries of MEDLE2 promoter/enhancer (if any) determined? It would have been wise to keep the genomic context of MEDLE2 by making an in-locus promoter fusion (i.e. replacing MEDLE2 ORF by a tagged version of MEDLE1). This would provide insight into whether MEDLE2 protein is better optimized for transport or its expression is strong enough to see accumulation in the cell.

– In Figure 6F, we would expect to see quantification of target gene expression under both uninfected and infected conditions.

*Reviewer #3:*

The manuscript demonstrates that MEDLE2 is exported by diverse parasite stages and fails to export when tethered to folded reporters. Targeted mutagenesis shows that Leucine 35 is required for the correct processing of MEDLE2, and is also a prerequisite for successful export. The authors go on to show that mammalian protein expression of MEDLE2 leads to an unfolded protein response and that this correlates fairly well with the response seen in vivo. The work is well done and the manuscript is clearly written. This paper opens up the field for exploration of effector proteins and their export mechanism.

– Brefeldin A experiments suggest that transit through the secretory pathway is required, though viability of the treated parasites needs to be demonstrated. A poison like azide would also block transit without demonstrating that vesicular transport is involved.

– Endogenous tagging of MEDLE2, as well as trials of knocking out the gene showed that the wild type locus is present in the transfected population even after the integration of the transgene cassette. Based on that, the author concluded that there are multiple copies of the MEDLE2 gene in *C. parvum* genome. However, might an alternative explanation be that the transfected population is a mixture of wild type and genetically modified parasites?

– The MEDLE2-GFP fusion that was used to determine the effect of the protein on host cells has 15 residues (aa 21 to 35) more than what would be expected to remain on the protein exported from the parasite. This is very likely not an issue, but the caveat should be noted.

– In figure 5D, it would be good to have both infected and uninfected cells to show the background fluorescence.

– Lines 93-94 refer to Figure 1B and C.

– Is there some parasite retention of the I36 mutant? Looks like there might be from the one example shown.

– The most highly upregulated gene is not identified by name, contrary to what is said in the text.

---

## [Author Response]

Reviewer #1:The host cell response could be explored in more details. There are several UPR pathways and it is not clear whether canonical UPR proteins are affected by MEDLE2. A previous study showed that C. parvum induces an ER response in HCT-8 cells (PMID 23986438), and this paper should be discussed here.

Thank you for bringing this to our attention. It was an oversight not to cite this paper. Morada et al.*,* 2013 showed *C. parvum* induces an ER stress response in HCT-8 cells, which was characterized by increases in stress response proteins (Calreticulin, GRP78, Nrf2, phosphoEIF2, and CHOP) by Western Blot. These findings align with our experimental observations of an ER stress response present both in the MEDLE2 transfection and *C. parvum* infection experiments (and we now test a CHOP KO mouse). We now specifically mention the findings of Morada et al., in the discussion (lines 373 -375).

The RNAseq data (Figure 6D) shows that only a small fraction of ER stress response genes are significantly upregulated, and there are other genes that are even more upregulated. What are these genes? A full list of DGE should be provided as a supplemental table and discussed in the text.

The full list of DGE were provided as in the Supplementary files submitted for both the MEDLE2 cell transfection dataset and the *C. parvum* infection dataset. We agree this data would be clearer if mentioned directly in the text instead of just providing the data in the supporting information. We have made this change as suggested and referenced a new table, Figure 6- source data 1. This table contains the full list of GSEA results with the NES, p value and gene names in the pathways. We also explained in more detail in the text how we narrowed in to study the ER stress response from the GSEA results through functional annotation clustering performed in DAVID on the full list of DEGs (lines 271-277).

A key question is whether the observed ER stress response is a byproduct of infection or is triggered by the parasite to promote its growth and/or survival. There are a number of simple experiments, based on knockout or knockdown of key UPR components, or using agonists or chemical inhibitors, which could provide elements of response.

Thank you for this suggestion. In response we have tested the impact of chemical inhibition of the UPR in HCT8 cells during *Cryptosporidium* infection using inhibitors. We chose inhibitors specific for 2 UPR signaling pathways for which we observed changes in our RNA sequencing datasets from MEDLE2 transfection and infection. We found that KIRA6, which targets the IRE1 branch of the UPR, reduces infection in vitro. Interestingly, GSK2606414 which inhibits the PERK signaling pathway had little impact on parasite growth. In support of these observations, infection of *Ddit3 -/-*mice (also n=known as CHOP, *n =* 12) resulted in reduced parasite burden when compared to infection of BL6 control mice (*n =* 12), which we interpret to suggest that the ER stress response does contribute to parasite survival. The results from both the in vitro inhibitor infections and *Ddit3-/-* mouse experiments have been added to Figure 6, as panels G and H, with corresponding figure legends added (Lines 1052-1065).

Another open question is the function of MEDLE2. Ectopic expression via transfection of MEDLE2 in host cells may not reflect the natural responses during infection. While the presence of multiple copies of MEDLE genes is probably a major obstacle for genetic studies, the authors could test whether the host cell response differs when MEDLE2 is exported (wildtype protein) or not (mutant versions). This would provide stronger evidence that in the context of infection MEDLE2 is directly responsible (at least in part) for the observed host cell response. Related to this point, line 28 and 75: there is no evidence that MEDLE2 is a pathogenesis factor

We agree that ablating the gene(s) would be the most direct avenue to understand their function. However, to date, this has not been possible due to multiple technical limitations. Genetic modifications that ablate export (DSP and L35A) were conducted using ectopic copies of MEDLE2, while the remaining endogenous cop(ies) remain intact and translocated to the host cell. We successfully engineered a parasite strain in which MEDLE2 was replaced with a marker (Figure 1—figure supplement 2); however, additional wild type copies remained. The recent reannotation of the *C. parvum* genome revealed MEDLE2 genes at multiple telomeric sites (Baptista et al., 2021) and we discuss this further in our response to reviewer 3. We did not observe an overt loss of fitness in ectopic mislocalization mutants that we engineered. They produce infections of similar parasite burden in culture and in animals.

We believe it likely that MEDLE proteins are pathogenesis factors, they are exported into the host cell, and they are under significant selection and encoded in complex highly malleable loci (see below), however, the reviewer is absolutely right that this has not been formally demonstrated. We have thus removed the phrase “pathogenesis factor” from lines 28 and 75 and replaced it with the neutral phrase “parasite proteins”.

MEDLE2 processing shares common features with processing in Plasmodium/Toxoplasma, where aspartyl proteases are involved. Did the authors try to block MEDLE2 export with aspartyl protease inhibitors?

We had attempted to block MEDLE2 export with aspartyl protease inhibitors synthesized for use in *Plasmodium falciparum* (WEHI916, WEHI842) and *Toxoplasma gondii* (WEHI586). None of the compounds tested fully ablated export (it is our understanding that this is true for Plasmodium as well), but treatment with WEHI842 resulted in accumulation of MEDLE2 and did impact parasite growth. We recognize several potential limitations to these experiments. The specificity and drug sensitivity of the *Cryptosporidium* enzyme may be different for that of these other parasites, as may be compound access due to difference in intracellular localization. For these reasons, we decided not to include this data in the current manuscript. We are conducting experiments that seek to explore genetically which *C. parvum* aspartyl protease is responsible for processing exported proteins; however, there are multiple candidates, and these are complex time-consuming experiments that we have not yet concluded, as some of the genes appear to be essential.

The image quality of Fig5D is not optimal, the MEDLE2-HA signal is very weak.

We apologize for the poor quality. We have improved the contrast and very much hope the reviewer finds it easier now to appreciate the staining. We cannot redo the full microscopy off all mutants and controls show here. Unfortunately, cryopreservation for *Cryptosporidium* oocysts is not fully robust yet. As a result, the parasites need to be passaged through mice to maintain viability every 6 months. Unfortunately, we had to triage what we could keep alive during the pandemic due to our limited access the lab and our mouse facilities.

Reviewer #2:– Lines 368-371: the authors state that "It remains to be determined whether the UPR is a byproduct of host detection of the disordered protein MEDLE2 following its translocation during C. parvum infection, or if it is an active process deliberately triggered to promote parasite survival." A key experiment to address this dilemma rightly formulated by the authors would be to express in 293 cells in parallel to MEDLE2, instead of GFP, a mock protein with an intrinsically disordered region (e.g. a Tg exported GRA protein), and monitor by qRT-PCR the expression levels of the genes mentioned in Figure 6F.

As suggested, we cloned TgGRA16 (aa 24-505) into the GFP expression plasmid and performed transfection of HEK293T cells in parallel with MEDLE2-GFP and the GFP only control. qPCR revealed changes in the ER stress response genes CHAC1, DDIT3, NUPR1, and TRIB3. We include these new experiments as Figure 6—figure supplement 2. The experiment shows that multiple unfolded parasite proteins can trigger the response upon transfection. It does not address whether *Cryptosporidium*, or *Toxoplasma* for that matter, benefits from that response to their exported unfolded proteins. We find reduced *Cryptosporidium* growth with pharmacological inhibition and genetic ablation experiments – but it’s not a complete block (see response to reviewer one). We agree with the reviewer, that divining pathogen intent is difficult, and thus have taken care in the discussion not to overextend our claim, and to focus on our findings and measurements.

– Ectopic expression of MEDLE1 at the TK locus (Figure 1—figure supplement 3E-F) under the control of a fragment of the MEDLE2 promoter only greatly enhances extra-vacuolar MEDLE1 secretion. How were the functional boundaries of MEDLE2 promoter/enhancer (if any) determined? It would have been wise to keep the genomic context of MEDLE2 by making an in-locus promoter fusion (i.e. replacing MEDLE2 ORF by a tagged version of MEDLE1). This would provide insight into whether MEDLE2 protein is better optimized for transport or its expression is strong enough to see accumulation in the cell.

Little is known about Cryptosporidium promoters, nothing about whether there are enhancers. Intergenic regions are usually small in this very compact genome, and in contrast to e.g. *T. gondii*, 5’UTRs are extremely short. For most genes the transcription initiation site appears to be in close proximity of the initiation codon used for translation. By now we have tested upstream regions of about ten different genes, and empirically found these small regions to be sufficient to drive expression of a reporter gene, and to reproduce the stage specificity and strength of the native locus (see. e.g. Tandel et al., 2019). We did not conduct mechanistic studies into what exactly defines promoters (*Cryptosporidium* is a rather poor model for such studies for numerous technical reasons). There are several published studies exploring this question using *Toxoplasma,* which found epigenetic marks and transcription factor binding sites in proximity of the transcription initiation site. We do have extensive transcriptional data and using multiple transcriptomic datasets we found the transcription initiation site of MEDLE2 to fall within a few bp of the start of the coding region.

Maybe more importantly, we have experimentally tested the presumptive promoter by using it to drive ectopic (*medle2*)MEDLE2-HA expression in the TK locus. We found expression timing and targeting to be indistinguishable from MEDLE2 tagged in the native locus. We used the same sequence and strategy to construct all of our point mutations with highly consistent results. And thus again used this strategy to engineer the chimeric (*medle2*)MEDLE1-HA (as so often, experiment shown early in the article, in real-life were done towards the end). In planning the point mutation studies, we actually considered performing “in locus” experiments similar to the one the reviewer discussed here. In the end we decided against this, however. The main reason was that *Cryptosporidium* genetic engineering relies exclusively on double stranded break repair by homologous recombination. Painful past experience taught us that even short stretches of sequence identity of 10-20 bp can suffice to enable unwanted cross overs (see e.g. Vinayak et al., 2020). As a result, when targeting a locus with identical or highly similar sequences, the repair DNA construct has to be extensively recodonized. Changing the primary sequence and codon usage comes with its own costs and caveats and may impact transcription and translation efficiency. We thus opted to use ectopic expression here. We have made this point more explicitly now.

– In Figure 6F, we would expect to see quantification of target gene expression under both uninfected and infected conditions.

We apologize for the confusion in how the data is being displayed. The graph in 6F is displayed as the fold change for each the genes of interest compared to the uninfected control. The left most panel represents the fold change of the uninfected, which is 1 for each of the genes being tested. Showing the fold change of the uninfected for each gene may make this clearer and we have updated the graph to reflect this now.

Reviewer #3:The manuscript demonstrates that MEDLE2 is exported by diverse parasite stages and fails to export when tethered to folded reporters. Targeted mutagenesis shows that Leucine 35 is required for the correct processing of MEDLE2, and is also a prerequisite for successful export. The authors go on to show that mammalian protein expression of MEDLE2 leads to an unfolded protein response and that this correlates fairly well with the response seen in vivo. The work is well done and the manuscript is clearly written. This paper opens up the field for exploration of effector proteins and their export mechanism.– Brefeldin A experiments suggest that transit through the secretory pathway is required, though viability of the treated parasites needs to be demonstrated. A poison like azide would also block transit without demonstrating that vesicular transport is involved.

We fully agree with the reviewer that pharmacological experiments have their caveats. BFA has more limited effects than azide, as it targets ER to Golgi trafficking and does not cause generalized cellular asphyxiation. However, it is likely to block parasite growth as they depend on BFA sensitive pathways to build secretory organelles required for invasion. For this reason, we also conducted genetic experiments that ablated the signal peptide, and these produced very similar results. We note the extensive short-term use of BFA to study protein trafficking in apicomplexan parasites over the last 20 years (e.g. Soldati et al., 1998 studying rhoptry protein maturation, or Wickham et al., 2001 studying the export of PFEMP1, Tonkin et al., 2006 studying apicoplast targeting). Not every pathway is immediately ablated by BFA toxicity. While targeting to e.g. the rhoptry, the food vacuole, or the host cell is susceptible to BFA, targeting of the apicoplast or the mitochondrion is not (a small number of specific apicoplast proteins does traffic via Golgi, and those are susceptible). We limited our experiments to a single merogony cycle and added BFA 3h following invasion with 7h incubation prior to fixation. Nonetheless, we now formally acknowledge that BFA is likely to inhibit long-term parasite growth in the segment that describes this experiment.

– Endogenous tagging of MEDLE2, as well as trials of knocking out the gene showed that the wild type locus is present in the transfected population even after the integration of the transgene cassette. Based on that, the author concluded that there are multiple copies of the MEDLE2 gene in C. parvum genome. However, might an alternative explanation be that the transfected population is a mixture of wild type and genetically modified parasites?

We considered this alternative, however, our mapping studies indicated that the locus was properly modified, but an additional copy (copies) of WT MEDLE2 persisted independently (Figure 1- Supplemental Figure2). We note that this was the same for all our experiments targeting this locus (even when we did not disrupt the gene). In experiments where we introduced an epitope tag or a fluorescent reporter, we could observe the presence of the cassette by following the protein(s) it encoded. All parasites uniformly expressed the endogenously tagged MEDLE2-HA arguing against heterogeneity due to transgene loss. Furthermore, while passaging transgenic oocysts through mice, we maintain selection with paromomycin. The MEDLE2-HA-tdNeon strain has been passaged through mice 8 times, and the presence of the transgene and WT bands remained unchanged. The subtelomeric locus that is home to MEDLE2 has long puzzled us, however, a recent genome reannotation effort by the Kissinger lab (Baptista et al., 2021) found numerous copies of the MEDLE gene family in multigene cassettes at different telomeres. This is not due to assembly artefacts as it is based on nanopore long read, single molecule sequencing. Please refer to Figure 6 in https://doi.org/10.1101/2021.01.29.428682 which shows complex expression sites with numerous copies of MEDLE genes, and the detailed discussion provided in that article. The overall conclusion was that the population of *C. parvum* parasites infecting an animal maintains multiple alternative telomeres among its individuals which contains members of multigene families with MEDLE being most prominent. These parasites engage in constant sexual recombination, and it appears that different members of the population may carry different cassettes. Note that the rest of the genome is very stable. Similar cassettes are known from antigenically varied pathogenesis factors in other organisms. We believe it likely that this organization reflects the overall importance of this gene family in host-parasite interaction, but more work is needed to fully understand this. The Baptista article and the complex telomere architecture is mentioned directly in the discussion now.

– The MEDLE2-GFP fusion that was used to determine the effect of the protein on host cells has 15 residues (aa 21 to 35) more than what would be expected to remain on the protein exported from the parasite. This is very likely not an issue, but the caveat should be noted.

Thank you for pointing this out. We now have made note of this in the text to ensure that the reader is aware of what exactly this construct includes: “we note that this protein contains 15 amino acids (aa 21 to 35) that are likely missing in parasite exported MEDLE2 due to N-terminal processing” (lines 262-264).

– In figure 5D, it would be good to have both infected and uninfected cells to show the background fluorescence.

We apologize that we did not include images of these controls. The images provided in Figure 5D are cropped to show only a single parasite containing infected cell. We now provided uncropped images for each of the representative cells shown in panel 5D to include uninfected cells to demonstrate the background fluorescence as a supplemental figure (Figure 5—figure supplement 3).

– Lines 93-94 refer to Figure 1B and C.

Thank you for this suggestion. We have incorporated this citation for figure 1B and C in the text (lines 93-94).

– Is there some parasite retention of the I36 mutant? Looks like there might be from the one example shown.

This is an astute observation by the reviewer, and we did not initially notice this ourselves. We do agree that the image provided appears to have accumulated MEDLE2 near the site of infection. We looked at additional images and believe the image here may represent MEDLE2 being exported within a normal context, as some images show MEDLE2 accumulation, while others do not. We cannot exclude the possibility that mutation of I36A may lead to accumulated MEDLE2 in the parasite; however, it does not ablate MEDLE2 export as is the observation for L35A.

– The most highly upregulated gene is not identified by name, contrary to what is said in the text.

We apologize for the confusion. To address this concern, we have added an additional source data file which includes a table (Figure 6-source data 1) that contains a full list of the GSEA results for the MEDLE2 transfection experiment. We also have provided a source data file which contains the core enrichment genes for both the transfection and infection datasets (Figure 6- source data 2). We also have added a statement about what most of the GSEA pathways involve and address how we narrowed in upon studying the ER stress response (Lines 273 -277).

References

Baptista, R.P., Li, Y., Sateriale, A., Sanders, M.J., Brooks, K.L., Tracey, A., Ansell, B.R.E., Jex, A.R., Cooper, G.W., Smith, E.D.*, et al.* (2021). Long-read assembly and comparative evidence-based reanalysis of *Cryptosporidium* genome sequences reveal new biological insights. bioRxiv, 2021.2001.2029.428682.

Tandel, J., English, E.D., Sateriale, A., Gullicksrud, J.A., Beiting, D.P., Sullivan, M.C., Pinkston, B., and Striepen, B. (2019). Life cycle progression and sexual development of the apicomplexan parasite Cryptosporidium parvum. Nat Microbiol *4*, 2226-2236.

Vinayak, S., Jumani, R.S., Miller, P., Hasan, M.M., McLeod, B.I., Tandel, J., Stebbins, E.E., Teixeira, J.E., Borrel, J., Gonse, A.*, et al.* (2020). Bicyclic azetidines kill the diarrheal pathogen *Cryptosporidium* in mice by inhibiting parasite phenylalanyl-tRNA synthetase. Science Translational Medicine *12*, eaba8412.a